# Atmosphere–ocean exchange of heavy metals and polycyclic aromatic hydrocarbons in the Russian Arctic Ocean

Xiaowen Ji[1,2], Evgeny Abakumov[2], Xianchuan Xie[1*]

[1] *State Key Laboratory of Pollution Control and Resource Reuse, Center for Hydrosciences Research, School of the Environment, Nanjing University, Nanjing 210093, P. R. China*

[2] *Department of Applied Ecology, Saint Petersburg State University, 16-line, 29, Vasilyevskiy Island, Saint Petersburg 199178, Russian Federation*

[*] Correspondence: Xianchuan Xie (xchxie@nju.edu.cn)

**Abstract.** Heavy metals and polycyclic aromatic hydrocarbons (PAHs) can greatly influence biotic activities and organic sources in the ocean. However, fluxes of these compounds as well as their fate, transport, and net input to the Arctic Ocean have not been thoroughly assessed. During April–November of the 2016 "Russian High Latitude Expedition", 51 air (gases, aerosols, wet deposition) and water samples were collected from the Russian Arctic within the Barents Sea, Kara Sea, Leptev Sea, and East Siberian Sea. Here, we report on the Russian Arctic assessment of the occurrence in dry and wet deposition of 35 PAHs and 9 metals (Pb, Cd, Cu, Co, Zn, Fe, Mn, Ni, and Hg), as well as the atmosphere–ocean fluxes of 35 PAHs and $Hg^0$. We observed that Hg was mainly in the gas phase and that Pb was most abundant in the gas phase compared with the aerosol and dissolved water phases. Mn, Fe, Pb, and Zn showed higher levels than the other metals in the three phases. The concentrations of PAHs in aerosols and the dissolved water phase were approximately one order of magnitude higher than those in the gas phase. The abundances of higher molecular weight PAHs were highest in the aerosols. Higher levels of both heavy metals and PAHs were observed in the Barents Sea, Kara Sea, and East Siberian Sea, which were close to areas with urban and industrial sites. Diagnostic ratios of phenanthrene/anthracene to fluoranthene/pyrene showed a pyrogenic source for the aerosols and gases, while the patterns for the dissolved water phase were indicative of both petrogenic and pyrogenic sources; pyrogenic sources were most prevalent in the Kara Sea and Leptev Sea. These differences between air and seawater reflect the different sources of PAHs through atmospheric transport, which included anthropogenic sources for gases and aerosols and mixtures of anthropogenic and biogenic sources along the continent in the Russian Arctic. The average dry deposition of $\sum_9$metals and $\sum_{35}$PAHs was 1749 ng m$^{-2}$ d$^{-1}$ and 1108 ng m$^{-2}$ d$^{-1}$, respectively. The average wet deposition of $\sum_9$metals and $\sum_{35}$PAHs was 33.29 μg m$^{-2}$ d$^{-1}$ and 221.31 μg m$^{-2}$ d$^{-1}$, respectively. For the atmosphere–sea exchange, the monthly atmospheric input of $\sum_{35}$PAHs was estimated at 1040 tons. The monthly atmospheric Hg input was approximately 530 tons. These additional inputs of hazardous compounds may be disturbing the biochemical cycles in the Arctic Ocean.

***Key words:*** Trace metals; PAHs; Russian Arctic; atmosphere–water fluxes

# 1 Introduction

The increasing anthropogenic activities associated with growing industries within boundary areas of the Arctic for economic reasons, including hydrocarbon exploration sites and mines in the Russian Arctic, represent potential pollution sources to Arctic ecosystems (Walker et al., 2003; Dahle et al., 2009; Ji et al., 2019). Additionally, the Arctic has long been contaminated by pollutants transported to polar areas from distant locations outside of this region (Hung et al., 2016). For example, anthropogenic sources of pollutants in the Arctic have been found to come from the Norilsk industrial area in the Taymyr Peninsula (Reimann et al., 1997; Zhulidov et al., 2011) and from the copper-nickel mining industry in the Kola Peninsula (Boyd et al., 2009; Jaffe et al., 1995). For pollutants transported from outside of the Arctic, reducing global emissions would be an ideal strategy to lessen the impacts of pollutants on Arctic ecosystems. For example, worldwide emissions of mercury will have increased by 25% in 2020 over 2005 levels according to previous estimations (Pacyna et al., 2010). Mercury is a key problematic pollutant in the Arctic because it is a neurotoxic pollutant significantly influencing northern latitudes through human exposure from eating seafood and marine mammals (Stow et al., 2015). Thus, global emission reductions could help to alleviate problems associated with long-range mercury transport and contamination in the Arctic. In regard to sources close to the Arctic, these may inevitably cause localized ecological risks or risks over a wider regional range. For instance, Fernandes and Sicre (1999) showed that atmospheric transport of anthropogenic polycyclic aromatic hydrocarbons (PAHs) to the Eurasian Arctic mainly originated from Eastern Europe and Russia. PAHs in aerosols from lower latitude were deposited on soils and ice in winter and transported by rivers to the ocean by the occurrence of freshet (Fernandes and Sicre, 1999). The previous study also showed a strong net deposition in the marine transect from East Asia to the Arctic, and the controlling sources both contained potential continental source region as East Asia and the influence of seasonal and regional source as forest fires in the Arctic (Ma et al., 2013). In addition, high concentrations of heavy metals (Mn, Zn, Ni, Fe and Cd) were observed in the west Arctic Ocean (Chukchi Sea); this enrichment was not only from Pacific-origin inflow water from the Bering Stat but also from additional sources such as melting sea ice and river water discharge (Kondo et al., 2016). Also of concern is that, with rapid warming of the global climate, the melting of contaminated ice may lead to more pollutant emission into the Arctic Ocean, which could harm its fragile ecosystems.

Pollutants can be transported to the Arctic through both seawater and atmospheric pathways; the atmospheric pathway is the quickest and most direct way for long-range pollutant transportation, e.g., pollutants can be transported from distant sources to the Arctic within several days or weeks (Shevchenko et al., 2003). Reports have revealed that some pollutants such as heavy metals and polycyclic aromatic hydrocarbons (PAHs) can be transported with aerosols over thousands of kilometers to Arctic regions (Rahn and Lowenthal, 1984; Maenhaut et

al., 1989; Shaw, 1991; Cheng et al., 1993). Approximately 100 tons of airborne mercury originating from industrial sources are deposited per year in the Arctic Ocean (Cone, 2008). While there is evidence that atmospheric inputs make large contributions to the chemical budgets in marine areas, the exact role of these inputs in the Arctic Ocean remains uncertain and may have been previously underestimated (Duce et al., 1991). Numerous studies have shown that aerosol transport is essential to transfer atmospheric compounds from air to ocean, and that this process is susceptible to changes in the climate of Arctic regions (Leck et al., 1996; Sirois and Barrie, 1999; Bigg and Leck, 2001). The compounds in aerosols over the Russian Arctic have been reported to show maximal concentrations during the winter/spring season; in addition, 50% of the air pollutants were found to have originated from Russian Arctic pollution (Shevchenko et al., 2003). It has also been reported that the natural biodegradation rates of exogenous compounds in the Arctic Ocean could be lower than those in more temperate oceans such as the Atlantic and Pacific (Bagi et al., 2014). In addition, Vieira et al. (2019) found that Fe, Mn, and Co were predominantly controlled by reductive benthic inputs, and that their levels were affected by the biological processes of uptake and release in the Arctic Ocean. Because of their toxicity and persistence, high concentrations of heavy metals or other persistent pollutants such as PAHs may disturb the benthic fluxes in cross-shelf mixing in Arctic regions, which could result in adverse effects on marine life and, with the eventual biomagnification in the food web, on humans as well. However, the long-term influence of heavy metals and PAHs on biogeochemical cycles in the Arctic Ocean remains poorly understood.

Atmosphere–seawater exchange is the main process that controls the residence time and levels of chemical compounds in the Arctic Ocean. In particular, atmospheric deposition is a significant source for pollutants in seawater, and dry deposition in the ocean has been widely studied (Jickells and Baker, 2019; Wang et al., 2019; Park et al., 2019). Although wet deposition (precipitation scavenging) is regarded as playing a predominant role in eliminating pollutants in both gas and particulate phases, current reports on the spatial distribution of pollutants from wet(snow) deposition in high-latitude oceans are scarce (Custódio et al., 2014). Moreover, for volatile or semivolatile compounds, the volatilization process is an important pathway for atmosphere–seawater exchanges. Therefore, the atmosphere–water exchange of volatile or semivolatile compounds can be estimated by the net flux of pollutants either volatilizing from seawater to air or depositing from air to seawater (Rasiq et al., 2019; Cheng et al., 2013; Totten et al., 2001). Gonzalez-Gaya et al. (2016) reported on a global assessment of atmosphere–ocean fluxes of 64 PAHs; the net atmospheric PAH input to global ocean was 0.09 Tg per month. The atmosphere–seawater exchange rate is greatly influenced by atmospheric temperature variations, and the direction and magnitude of fluxes of compounds between air and seawater vary seasonally (Bamford et al., 1999; Hornbuckle et al., 1994). Additionally, inorganic salt ions can decrease the aqueous solubility of organic compounds such as PAHs

(Rasiq et al., 2019). During melting of sea ice in the Arctic Ocean, the magnitude and direction of atmosphere–seawater fluxes may be different from those in tropical and subtropical oceans (Gonzalez-Gaya et al., 2016; Rasiq et al., 2019). The Arctic Ocean is considered as a sink that receives global airborne pollutants (Environment-Canada; et al., 2008); however, the fate of atmosphere–ocean exchange of trace metals and organic compounds remain unclear.

In this study, two categories of pollutants (i.e., 9 heavy metals and 35 PAHs) were measured in the Arctic Ocean, in aerosols, gas, and seawater, and atmosphere–ocean exchanges of Hg and PAHs was studied. We hypothesized about the relative equilibrium of chemical exchanges between seawater and air and calculated the net diffusion of atmosphere–ocean exchange of Hg and PAHs in the Arctic Ocean for an evaluation of the double-directional exchange. Meanwhile, the dry and wet deposition of heavy metals and PAHs in the Russian Arctic Ocean were determined. The distributions of heavy metals and PAHs in each sea of the Arctic Ocean and in various phases were also characterized to identify possible sources from the continents.

## 2 Materials and methods

### 2.1 Study area and sample collection

All samples were collected during the period of April 9 to November 10, 2016 as part of the "Russian High Latitude Expedition" carried out on the vessel *Mikhail Somov* (this vessel traveled from the city of Arkhangelsk to Wrangel Island). Fifty-one air and water samples, and eight wet deposition samples were gathered from locations ranging from the southern inlet of the Barents Sea (from west sites to Vayach Island) to across the Kara Sea (to Gerkules Island), Laptev Sea (to Bennett Island), and East Siberian Sea (to Wrangel Island) (**Fig. 1**).

### 2.1.1 Aerosol and gas phase

Air samples, including aerosols and concurrent gases as described elsewhere (Reddy et al., 2012; Shoeib and Harner, 2002; Galarneau et al., 2017; Grosjean, 1983; Wu, 2014), were collected by a high-volume sampler set up at the top of a main rod. A wind vane was connected to the high-volume sampler so that samples could be collected only if the wind was derived from the bow to prevent contamination from ship emissions. The average sampled air volume was 632 m$^3$ (412–963 m$^3$) per sample. The aerosols were sampled on Teflon filters (P0325-100EA, Fluoropore, Darmstadt, Germany), and then, the compounds in the gas phase were collected over precleaned polyurethane foams (PUFs). After sampling, the filters and PUFs were tightly covered with aluminum foil for air-tightness, then immediately placed in polyethylene bags, and frozen at -20 °C prior to chemical analyses.

### 2.1.2 Wet deposition and water

Wet deposition samples were collected through a cleaned stainless steel funnel connected to a glass bottle during eight snow events. Snowfall samples were melted thoroughly at room temperature. Water samples were gathered continuously from surface seawater (5 m depth) along the vessel, and these samples were immediately filtered onto borosilicate microfiber glass filters (AP1504700, EMD Millipore, Darmstadt, Germany). Then, the compounds in the dissolved phase were retained on XAD sorbent tubes subjected to controlled flows. The mean

filtered water volume was 1239 mL (135–2876 mL). The XAD tubes were stored at 5 °C before their extraction in the laboratory.

**2.2 Heavy metal extraction and analysis**

For metal determinations in the aerosol, gas phase, wet deposition, and water samples, Teflon filters, PUFs, and dissolved phases were first Soxhlet-extracted for 8 h by using $HNO_3$. The samples were then diluted with

deionized water to 23 mL and subjected to inductively coupled plasma mass spectrometry (ICP-MS) analysis. Specifically, the contents of Pb, Cd, Cu, Co, Zn, Fe, Mn, Ni, and Hg were analyzed on an ICP-MS instrument (Thermo Scientific ICE 3500, Waltham, MA, USA) while making use of rhodium (Rh) as an internal standard. High-resolution (10,000) data were collected to avoid any mass interference problems.

**2.3 PAH extraction and analysis**

For PAH determinations in the gas, aerosol, and dissolved phase samples, published procedures were used (Berrojalbiz et al., 2011; Castro-Jimenez et al., 2012; Gonzalez-Gaya et al., 2014). Snow-melt water was extracted by using solid phase HLB Oasis cartridges (60 mg/3 cc) on board. Briefly, cartridges were preconditioned with 5 mL methanol, 10 mL of a mixture of methanol:dichloromethane (1:2), and 10 mL deionized water. Afterward, each sample was combined with a recovery standard and concentrated by $N_2$ until near dryness. Then, it was eluted with

5 mL hexane, 5 mL of a mixture of hexane:dichloromethane (1:2), and 10 mL deionized water.

Thirty-five PAH species were quantified, including naphthalene, methylnaphthalene (sum of two isomers), 1,4,5-trimethylnaphthalene, 1,2,5,6-tetramethylnaphthalene, acenaphthylene, acenaphthene, fluorene, dibenzothiophene, anthracene, 9-methlyfluorene, 1,7-dimethylfluorene, 9-n-propylfluorene, 2-methyldibenzothiophene, 2,4-dimethyldibenzothiophene, 2,4,7-trimethyldibenzothiophene, 3-methylphenanthrene,

1,6-dimethylphenanthrene, 1,2,9-trimethylphenanthrene, 1,2,6,9-tetramethylphenanthrene, fluoranthene, pyrene, benzo[*a*]anthracene, chrysene, 3-methylchrysene, 6-ethylchrysene, 1,3,6-trimethylchrysene, benzo[*b*]fluoranthene, benzo[*k*]fluoranthene, benzo[*a*]pyrene, perylene, dibenzo[*a,h*]anthracene, indeno[*1,2,3-cd*]pyrene, dibenzo[*a,h*]anthracene, and benzo[*g,h,i*]perylene. PAH quantification was performed by gas chromatography-mass spectrometry (GC-MS). Specifically, we used a gas chromatograph coupled with a triple quadrupole mass

selective detector (GS-MS, ITQ 1100, Thermo Scientific, USA) equipped with a DB-5MS chromatographic capillary column (30 m × 0.25 mm i.d. and 0.25-μm film, Agilent Technologies, Santa Clara, CA, USA) operating in electron impact mode (EI) and with selected ion monitoring (SIM) as reported previously (Gonzalez-Gaya et al., 2014). Internal standards (anthracene-$d_{10}$, $p$-terphenyl-$d_{14}$, pyrene-$d_{10}$, and benzo[$b$]fluoranthene-d12) were added before operating the GC-MS instrument for the quantification of PAHs, and the recovery of perdeuterated standards (acenaphthene-$d_{10}$, chrysene-$d_{12}$, phenanthrene-$d_{10}$, and perylene-$d_{12}$) was determined by addition prior to the procedures of extraction; these values were then used for the correction of measured concentrations.

## 2.4 Quality assurance and quality control

Analyses of every sample and phase were conducted in the laboratory with field blanks to determine the analytical limits and recoveries. Breakthroughs of aerosols and gas phases were checked for the Teflon filter and PUF samples. Approximately 90% of the metals and PAHs were obtained during the first half of the sample analysis, while the remaining 10% were obtained during the second half; for the PAHs, these mostly consisted of compounds with 2–3 rings. Six blanks (field and laboratory) were collected for the gas phase, while seven field banks and eight laboratory blanks were used for the dissolved phase, all of which were extracted along with the rest of the samples during the analytical procedure. For the gas phase, average $\sum$metal values were approximately 0.049 and 0.052 ng per sample in the field and laboratory blanks, respectively, and average $\sum$PAH values were approximately 2.44 and 2.06 ng per sample in the field and laboratory blanks, respectively (**Table S1 and S2, Supplementary material**). For the aerosols, average $\sum$metal values were 0.046 and 0.065 ng per sample in the field and laboratory blanks, respectively, and average $\sum$PAH values were 2.95 and 2.96 ng per sample in the field and laboratory blanks, respectively. Likewise, for the dissolved phase, values of 0.053 and 0.052 ng per sample were obtained for the $\sum$metals and values of 2 and 1.73 ng per sample were obtained for the $\sum$PAHs. All measured PAHs from field samples exceeded the field and laboratory blank concentrations; therefore, the quantified compounds did not subtract the blank values. Mean recoveries of perdeuterated standards used as surrogates in dissolved samples were as follows: 63% for acenaphthene-$d_{10}$, 54% for chrysene-$d_{12}$, 73% for phenanthrene-$d_{10}$, and 82% for perylene-$d_{12}$.

All concentrations in each medium were corrected by the surrogate recovery for individual samples. The detection limit was used for the lowest limit of the calibration curve. The quantification limit was equivalent to the average blank concentration for each phase.

## 2.5 Data processing

Dry deposition fluxes ($F_{DD}$, ng m$^{-2}$ d$^{-1}$) were calculated from field measurements of trace metals and PAHs collected during the expedition over eight time periods. Aerosol deposition fluxes for the metals were calculated as

follows:

$$F_{DD}(\text{metal}) = C_d V_d, \tag{1}$$

where $C_d$ is the concentration of atmospheric aerosols and $V_d$ is the velocity of deposition (m s$^{-1}$). $V_d$ was calculated as shown in Eq. (2), and the details have been described elsewhere (Zhang et al., 2001):

$$V_d = u_{grav} + \frac{1}{R_a + R_s}, \tag{2}$$

where $u_{grav}$ is the gravitational settling velocity and $R_a$ and $R_s$ are the aerodynamic resistance for gaseous species and the surface resistance, respectively. $R_s$ can be calculated as follows:

$$R_s = \frac{1}{\varepsilon_0 u_*(E_B E_{IM}) R_1}, \tag{3}$$

where $\varepsilon_0$ is an empirical constant ($\varepsilon_0 = 3$) and $u_*$ is the friction velocity calculated for gases. $E_B$ is the collection efficiency of Brownian diffusion as a function of the Schmidt number Sc:

$$E_B = (Sc)^{-\gamma}, \tag{4}$$

where $\gamma$ is an empirical constant ($\gamma = 0.5$). $E_{IM}$ is the collection efficiency from impaction based on the following formulas (Peters and Eiden, 1992):

$$E_{IM} = (\frac{st}{0.8 + st})^2, \tag{5}$$

$$st = \frac{V_{grav} V_*{}^2}{V_a}, \tag{6}$$

where $V_a$ is the kinematic viscosity for air (m$^2$ s$^{-1}$). The correction factor ($R_1$) is the fraction of particles close to the surface:

$$R_1 = \exp{(-st^{1/2})} \tag{7}$$

For PAHs, the specific compound deposition velocity ($V_d$, cm s$^{-1}$) was derived from an empirical parameterization (Gonzalez-Gaya et al., 2014):

$$\log(V_d) = -0.261 \log(P_L) + 0.387 U_{10} Chl_s - 3.082, \tag{8}$$

where $P_L$ is the subcooled liquid vapor pressure of each PAH, $U_{10}$ is the 10 m height wind speed, and $CHl_s$ is the concentration of surface chlorophyll. With Eq. (8), one can estimate the $V_d$ for each PAH and sampling period by taking $P_L$ from references and using the field-measured $U_{10}$ and $CHl_s$. In this study, $F_{DD}$(PAH) values were estimated from the measured concentrations in the aerosol phase ($C_A$, ng m$^{-3}$) as follows:

$$F_{DD}(PAHs) = 864 V_d C_A, \tag{9}$$

where 864 is the unit conversion factor.

The wet deposition fluxes ($F_W$, ng m$^{-3}$ d$^{-1}$) of metals/PAHs were estimated by using the quantified concentrations of metals/PAHs from the collected snow and the precipitated volume of snow-melt water per surface and time period for each of the eight snow events during the expedition.

The air–water diffusive fluxes ($F_{AW}$, ng m$^{-2}$ d$^{-1}$) for Hg/PAHs were calculated according to Fick's law:

$$F_{AW} = K_{AW}\left(\frac{C_G}{H'} - 1000 C_{TW}\right), \tag{10}$$

where $C_G$ and $C_{TW}$ represent the concentration measured in the gas phase (ng m$^{-3}$) and dissolved phase (ng L$^{-1}$), respectively. $H'$ is the temperature dependence of Henry's law constant, and $H'$ values for PAHs were taken from Bamford et al. (1999); $H'$ for Hg was calculated from Eq. (11) for seawater (Andersson et al., 2008):

$$H' = \exp\left(\frac{-2404.3}{T} + 6.92\right), \tag{11}$$

where T is the temperature of the surface water (K). $H'$ was corrected by the field-measured salinity. $K_{AW}$ represents the air–water mass transfer rate (m d$^{-1}$) calculated by a two-film model (Singh and Xu, 1997) and while considering the nonlinear wind-speed effect. $C_{TW}$ for Hg was directly measured concentrations, and $C_{TW}$ values for PAHs were calculated by using the measured concentrations in the dissolved phase ($C_W$) as follows:

$$C_T = \left(\frac{C_W}{1 + k_{DOC} DOC}\right), \tag{12}$$

where $K_{DOC}$ was taken as the value of 10% of the octanol–water partitioning coefficient ($K_{OW}$) (Burkhard, 2000), and DOC represents the dissolved organic carbon (mg L$^{-1}$).

$K_{AW}$ was calculated by the two-film model:

$$\frac{1}{K_{AW}} = \frac{1}{K_W} \frac{1}{K_A H'}, \tag{13}$$

where $K_W$ and $K_A$ are the mass transfer coefficients (m d$^{-1}$) of Hg and PAHs in the water and air films, respectively. The mass transfer coefficient of $CO_2$ in the water phase ($K_{W,CO_2}$, m d$^{-1}$) can be used to calculate $K_W$ (Gonzalez-Gaya et al., 2016), which is a wind-speed quadratic function at a height of 10 m ($U_{10}$, m$^{-1}$) (Nightingale et al., 2000). A Weibull distribution of wind speed was assumed to parameterize $K_{W,CO_2}$ because average wind speeds were used during the sampling period since the gas and dissolved phase concentrations were averaged values for the sampling transects; $K_{W,CO_2}$ was calculated by using a previously reported method (Livingstone and Imboden, 1993):

$$K_{W,CO_2} = 0.24\left[0.24\eta^2\Gamma\left(1 + \frac{2}{\xi}\right) + 0.061\eta\Gamma(1 + 1/\xi)\right], \tag{14}$$

where $\eta$ and $\xi$ are the constants of scale and shape in the Weibull distribution, respectively, and $\Gamma$ represents a gamma function. $\xi = 2$ (Rayleigh distribution) was used as recommended (Gonzalez-Gaya et al., 2016). $\eta$ is related to wind speed and was calculated with $U_{10} = \eta\Gamma(1 + 1/\xi)$ (Livingstone and Imboden, 1993).

$K_W$ can be calculated as follows:

$$K_W = K_{W,CO_2} \frac{1}{\sqrt{\frac{SC_{PAH}}{600}}}, \tag{15}$$

where $SC_{PAH}$ is the Hg/PAH Schmidt number. The same applies for $K_A$, which was also calculated from wind

speed and the $H_2O$ mass transfer coefficient for the air phase ($K_{A,H_2O}$, cm s⁻¹):

$$K_{A,H_2O} = 0.2U_{10} + 0.3, \tag{16}$$

$$K_A = 864K_{A,H_2O}\sqrt{\frac{D_{i,a}}{D_{H_2O,a}}}, \tag{17}$$

where $D_{i,a}$ and $D_{H_2O,a}$ represent the Hg/PAH and $H_2O$ diffusive coefficients in air, respectively.

The uncertainty was lower than a factor of one to two in these estimates for metals/PAHs. Most of the increasing uncertainty was associated with the Henry's law constants. The effect of uncertainty on the air–water exchange net direction was assessed by the ratios of air–water fugacity ($f_G$/$f_W$) (**Figs. S1 and S2, Supplementary material**); moreover, and the findings revealed that most metals and PAHs were not close to the equilibrium of air–water. Among the PAHs, net volatilization was detected only for dibenzothiophene, alkylated phenanthrenes, and fluoranthene. The details of the uncertainty analysis are shown in **Text S1** (**Supplementary material**).

Gross fluxes of volatilization and absorption depend on the first and second term of Eq. (10). The total accumulated fluxes for the Barents Sea, Kara Sea, Laptev Sea, and East Siberian Sea were acquired by multiplying the mean basin flux with its standard deviation by the surface area of each basin.

The estimations of degradation fluxes of PAHs in the atmospheric ocean boundary were calculated as follows:

$$D_{atm} = \frac{(C_{Gf} - C_{Gi})ABL}{t}, \tag{18}$$

where $C_{Gf}$ and $C_{Gi}$ are the last concentration after a fixed time in a closed system (ng m⁻³) and the concentration in the gas phase at the initial time (ng m⁻³), respectively. $t$ is the time period (average 5 h daytime per day), and ABL represents the average height of the atmospheric boundary layer (380 m). $C_{Gf}$ can be calculated as follows:

$$Ln(\frac{C_{Gi}}{C_{Gf}} = k_{OH}[OH]t) \tag{19}$$

where $k_{OH}$ is the rate constant for a PAH reaction with OH radicals (Keyte et al., 2013) and [OH] is the hydroxyl radical concentration in the mixed layer (1000–500 hPa) based on the monthly mean OH radical concentration (Spivakovsky et al., 2000). The mean concentrations of OH were calculated by Eq. (13). The OH concentrations ranged between 5.23 and $17.26 \times 10^5$ mol cm⁻³. Besides, only the PAHs in the gas phase were considered while the potential degradation of PAHs bound in aerosols was ignored. Considering the uncertainty of those sources, a relevant error factor of 2-3 was given for the degradative fluxes based on the individual PAHs. Because of the large uncertainties in $k_{OH}$ values, the degradation fluxes of PAHs in the atmosphere could not be calculated.

# 3 Results and discussion

## 3.1 Heavy metals in the atmosphere-ocean

Nine heavy metals were measured, and the average concentration for each metal in each sea can be found in

**Table S3**. The highest $\sum_9$ metal concentrations in the Barents Sea were found in the gas phase ($C_G$, ng m$^{-3}$), where the average concentration was 0.418 ng m$^{-3}$ (**Fig. S3**). The average values of $C_G$ showed no obvious differences among the four seas, whereas the oceanic area adjacent to the Chukchi Peninsula, Taymyr-Gydan Peninsula, and Arkhangelsk region showed higher combined concentrations of the nine metals (**Fig. 2a**). High $\sum_9$ metal concentrations in the aerosol phase ($C_A$, ng m$^{-3}$) were observed in the Barents Sea ($p < 0.05$), where the average $\sum_9$ metal concentration was 2.713 ng m$^{-3}$ (**Fig. 2b**). These high levels may have been associated with the trajectories of air from the Russian inlands. The distributions of heavy metals in the Russian Arctic Ocean revealed that the concentrations of the $\sum_9$ metals in seawater were lower than those in air. The concentrations of each metal in aerosols were comparable to those previously reported in the Russian Arctic, i.e., Leptev Sea, Kara Sea, Barents Sea, Sevenaya Zemlya, and Wrangel Island (Shevchenko et al., 2003; Vinogradova and Ivanova, 2017). For other parts of the Arctic Ocean, the average mass concentrations of each metal from Svalbard, the Fram Strait, Central Arctic, and Greenland (Ferrero et al., 2019; Maenhaut et al., 1979; Maenhaut et al., 2002; Maenhaut et al., 1989), were higher than those found in aerosols in our study. Metals' concentrations in aerosols in our study are lower than those in the Red Sea and Mediterranean area (Chen et al., 2008). The average $\sum_9$ metal concentrations in dissolved water ($C_W$) ranged from 0.526 to 0.896 µg L$^{-1}$ (Leptev Sea to Barents Sea). This is relatively lower than the concentrations of dissolved trace metals (Mn, Fe, Ni, Zn and Cd) previously reported for the western Arctic Ocean (Chukchi Sea and Canada Basin, depth: 5–20 cm) (Kondo et al., 2016). Higher values of $C_W$ were observed in the Barents Sea–Kara Sea region (Yamal Peninsula) and in the East Siberian Sea (close to the Chukchi Peninsula) in comparison to the $C_W$ values in other areas (**Fig. 2c**).

The abundance of each metal in gases, aerosols, and dissolved water is dependent on the emission sources. In this study, Fe and Zn were the most abundant metals detected in aerosols and dissolved water from the Russian Arctic Ocean, where the average $\sum_9$ metal concentrations in aerosols and dissolved water were 0.64 ng m$^{-3}$ and 0.91 ng L$^{-1}$, respectively. Pb was the most abundant metal in the gas phase (the average concentration in the Russian Arctic Ocean = 0.14 ng m$^{-3}$). In comparison to aerosols and dissolved water, the gas phase contained higher levels of Hg, which is a finding consistent with the usual form of Hg in the atmosphere (>98%) and the tendency for the remaining types of Hg to adsorb to particles during atmospheric transport (Poissant et al., 2008). In all phases, the proportions of Mn, Fe, Pb, and Zn were significantly higher than those of other heavy metals. Additionally, the metal distributions in the Barents Sea and Kara Sea showed the highest proportions, followed by the metal distributions in the East Siberian Sea. On the Taymyr Peninsula (adjacent to the Kara Sea and Leptev Sea), there is a mining and metallurgical factory operated by the company Norilsk that processes copper and nickel and is one of the biggest metallurgical factories in the world. This may be a likely source of metals in the Kara Sea region

(Shevchenko et al., 2003). Because of the significant differences in the concentrations of metals in the marine boundary layer both temporally and spatially throughout the Russian Arctic Ocean (Vinogradova and Polissar, 1995; Shevchenko et al., 1999), as well as the scarcity of reported data on heavy metals in the atmosphere in this region, it was difficult to compare our data with historical findings. However, our data are similar to those reported for September 1993 in the Kara Sea (Rovinsky et al., 1995).

The dry deposition that involves aerosols binding to heavy metals ($F_{DD}$, ng m$^{-2}$ d$^{-1}$) is a major process for heavy metal deposition (Shevchenko et al., 2003). In the Russian Arctic Ocean, the average $F_{DD}$ of the $\sum_9$ metals ranged from 392 to 8067 ng m$^{-2}$ d$^{-1}$ (mean = 1792 ng m$^{-2}$ d$^{-1}$). The largest $F_{DD}$ value was found close to the coast of the East Siberian Sea, where $F_{DD}$ values of 305 and 224 ng m$^{-2}$ d$^{-1}$ were observed for Hg and Pb, respectively, and were dominant (**Fig. 3a**). Our results seem to be one magnitude higher than those in the Red Sea (mean = 615 ng m$^{-2}$ d$^{-1}$) (Chen et al., 2008) and Mediterranean Sea (mean = 264 ng m$^{-2}$ d$^{-1}$; Chen et al. 2008). However, this comparison may not reflect the strength of the emission sources because dry deposition is highly dependent on the deposition velocity, which is affected by meteorological conditions such as the humidity, wind speed, and stability of the air column (Mariraj Mohan, 2016). The relative humidity in the Arctic Ocean tends to be higher in coastal areas and notably we sampled during spring–winter when water vapor evaporates from the relatively warmer surfaces of seawater (Vihma et al., 2008). In addition, the wind over sampling sites in the Arctic Ocean was ~ 7 m s$^{-1}$ on average (the largest average wind speed was ~ 9 m s$^{-1}$ in the Barents Sea), which is significantly higher than the wind in the Red Sea and Mediterranean Sea (0.36–1 m s$^{-1}$) (Chen et al., 2008; Chester et al., 1999). During the eight snow events encountered during the expedition, the wet deposition flux of the $\sum_9$ metals ($F_{WD}$, μg m$^{-2}$ d$^{-1}$) ranged from 23 to 32 μg m$^{-2}$ d$^{-1}$ (mean = 26 μg m$^{-2}$ d$^{-1}$) (**Fig. 4a**). Data relevant to the wet deposition flux of heavy metals in the Arctic region include results for Hg, which were estimated on land in Alaska; the highest deposition was detected along the southern and southeastern coasts (> 0.05 μg m$^{-2}$ d$^{-1}$) (Pearson et al., 2019). The values were quite similar to the $F_{WD}$ for Hg in our study (0.05 to 0.09 μg m$^{-2}$ d$^{-1}$). Through analysis of variance tests, we did not find any significant difference in the $F_{WD}$ at different locations ($p > 0.05$) for all heavy metals, while a relatively higher $F_{WD}$ for Hg was observed in coastal areas adjacent to the Taymyr Peninsula with industrial factories. Pearson et al. (2019) pointed out that there are larger contributions from Hg wet deposition in the Bering Sea and Gulf of Alaska, which are influenced by the western Pacific downwinds of East Asia, where high Hg emissions from industrial activities and coal burning occur (Wong et al., 2006). The Russian Arctic Ocean is also affected by Pacific downwinds, which could lead to a combination of heavy metal deposition from both local anthropogenic sources and long-range transport from Asia. Wet deposition is an important process for the transfer of heavy metals from gas and aerosol phases to ocean water. Snowfall in the Arctic is an important fraction of precipitation, but variations

in measurements ranging from 20% to 50% can occur under windy conditions even with sampling equipment designed with wind protection (Rasmussen et al., 2012). However, snow events are quite sporadic in the Russian Arctic Ocean during spring–summer compared with the other deposition processes. Nevertheless, wet deposition in our study was under regional influences and had a relatively high uncertainty.

For many heavy metals that form volatile species, there is additional evidence that their existence in water is strongly related to releases from terrestrial environments rather than internal cycling in aquatic systems (Robert, 2013). For example, following the deposition of atmospheric Fe, a non-volatile species, the concentrations in water are influenced mainly by the particulate phase and its dissolution, whereas for Hg, a volatile species that predominantly exists in the atmosphere as a gas ($Hg^0$), the concentrations of volatile Hg species in water are largely influenced by volatilization and deposition processes at the air–water interface; portions of the Hg in aquatic systems ends up being converted to methylmercury (Mason and Sheu, 2002; Sunderland and Mason, 2007; Selin et al., 2007; Strode et al., 2007). Hg concentrations in the gas phase in the present study were significantly lower than those measured in 1996 in the Northern Hemisphere (1.5–1.7 ng $m^{-3}$) and Southern Hemisphere (1.2–1.3 ng $m^{-3}$) (Steffen et al., 2005; Slemr et al., 2003; Wängberg et al., 2007; Kim et al., 2005). Steffen et al. (2002) indicated that there has been increasing retention of Hg in the Arctic region based on analyses of long-term measurements of atmospheric Hg concentrations. Diffusive air–water exchange is the dominant process driving the exchange of Hg in the ocean. The net diffusive air–water exchange ($F_{AW}$, ng $m^{-2}$ $d^{-1}$) was estimated by a two-film resistance model (Robert, 2013). The net input of Hg was calculated as shown in **Fig. 3b** and revealed that there was a net input from the atmosphere to the ocean at all stations, especially for the stations close to industrial/urban areas. The integrated monthly $F_{AW}$ fluxes (tons per month) for Hg were of the same order of magnitude as the $F_{DD}$ fluxes for Hg and other heavy metals in the Russian Arctic Ocean (**Fig. S4**). For Hg, the gross volatilization and gross absorption in the Russian Arctic Ocean were 250 and 530 tons per month, respectively. In consideration of previous studies of atmospheric mercury depletion events (AMDEs), during which the net input of Hg in the Arctic was evaluated (Brooks et al., 2006; Lindberg et al., 2001), we adjusted our sampling times to avoid sampling during sunrise when the autocatalytic release of sea salt aerosols changes the oxidative photochemistry in the stratified planetary boundary layer where elemental and reactive Hg in the gas phase is oxidized by reactive halogens. It was estimated that ~ 99-496 tons of Hg are deposited per year in the Arctic during AMDEs (Skov et al., 2006; Ariya et al., 2004). The net input of Hg in the present study was one order of magnitude higher than that caused by the AMDEs; this discrepancy may result from the fact that the previous studies considered the terrestrial boundary between air masses and snowpack, and that the different locations and seasons were affected by different meteorological conditions. In northern regions, it has been shown that Hg undergoes long-range transport from Eurasia, especially

during the winter season (Poissant et al., 2008). These net amounts of Hg entering into the Arctic Ocean pose potential risks to marine biota because Hg is poorly mobile and can be retained by aquatic biota that are exposed to it during the deposition process (Harris et al., 2007).

**3.2 PAHs in the atmosphere-ocean**

Thirty-five individual PAHs, which included isomer groups such as alkylated PAHs, were measured. The average concentrations of PAHs in each sea of the Russian Arctic Ocean are shown in **Table S4**. The average values of $C_G$ showed no obvious differences in the Kara Sea, Laptev Sea, and East Siberian Sea ($p > 0.05$), and no particularly high levels of $C_G$ were detected at all of the sampling sites (**Fig. 5a**). The range of $\sum_{35}$ PAH $C_G$ is 19.87–22.14 ng m$^{-3}$ for the Barents Sea, 19.01–22.34 ng m$^{-3}$ for the Kara Sea, 19.23–-21.70 ng m$^{-3}$ for the Leptev Sea, and 19.28–22.61 ng m$^{-3}$ for the East Siberian Sea. The highest $C_G$ of $\sum_{35}$ PAH is observed in the Barents Sea, with a value of 22.61 ng m$^{-3}$. $\sum_{35}$ PAH $C_A$ in the Barents Sea (0.25–2.95 ng m$^{-3}$), and East Siberian Sea (0.24–3.32), with average values of 1.38 and 2.07 ng m$^{-3}$ , respectively, were higher than those in the Leptev Sea (0.23–0.89 ng m$^{-3}$) and Kara Sea (0.23–0.27 ng m$^{-3}$), with average values of 0.30 and 0.25 ng m$^{-3}$, respectively (**Fig. 5b**). The average $C_A$ of $\sum_{35}$ PAH in the present study is higher than the average $C_A$ of $\sum_{64}$ PAH measured in the South Atlantic Ocean (0.93 ng m$^{-3}$) and North Pacific Ocean (0.56 ng m$^{-3}$), while much lower than the average $C_A$ of $\sum_{64}$ PAH in the Indian Ocean (10 ng m$^{-3}$) (Gonzalez-Gaya et al., 2016). The average $\sum_{35}$ PAH $C_A$ of 1.02 ng m$^{-3}$ in the Russian Arctic Ocean is comparable to the average $\sum_{64}$ PAH $C_A$ observed in the South Atlantic Ocean and South Pacific Ocean (1.1 ng m$^{-3}$) (Gonzalez-Gaya et al., 2016). $\sum_{18}$ PAH $C_A$ was measured from the North Pacific towards the Arctic Ocean, ranging from 0.0002 to 0.36 ng m$^{-3}$, with the highest concentration found in the coastal areas in East Asia (Ma et al., 2013). These concentrations were significantly lower than the average levels found in our study. Besides, Ma et al. (2013) observed the relatively higher $\sum_{18}$ PAH $C_A$ in the most northern latitudes of the Arctic Ocean, which is associated with back trajectories of air masses from Sothern Asia. The higher levels of $C_A$ in our study could be attributed to the costal line being close to larger areas of burning taiga forest and more industrial sources in the boreal regions of Russian continent. Similar to the pattern for heavy metals mentioned above, high levels of these chemicals may have been derived from atmospheric transport from the industrial areas of the Russian continent. Because of the various sampling methods and, the differences in PAHs measured, and because not all studies separated gas and particles concentrations, it is quite difficult to compare PAH levels in aerosols. The average $\sum_{35}$ PAH $C_W$ ranged from 13.07 ng L$^{-1}$ (Laptev Sea) to 69.90 ng L$^{-1}$ (Barents Sea), and its spatial variability was similar to that of PAH concentrations in aerosols. The range of $\sum_{35}$ PAH $C_W$ for the Barents Sea, Kara Sea, Leptev Sea, and East Siberian Sea is 12.36–162.05 ng L$^{-1}$, 12.18–14.04 ng L$^{-1}$, 11.21–15.82 ng L$^{-1}$, and 11.40–129.60 ng L$^{-1}$, respectively. Higher levels of $C_W$ were also found along the coast of the Yamal-Gydan

Peninsula, where petrol and natural gas industries have active sites (**Fig. 5c**).

The contribution of each PAH in the gas, aerosol, and dissolved water phases is determined by its source, volatility, and hydrophobicity (Lima et al., 2005). The low-molecular weight PAHs were dominant in gas and dissolved water (**Fig. S5**). In the gas phase, low-molecular-weight PAHs occupied more than 75% of the $\sum_{35}$ PAHs, which mainly contained methylated phenanthrenes, e.g., methylphenanthrene (mean = 1.31 ng m$^{-3}$), dimethylphenanthrene (mean = 1.27 ng m$^{-3}$), and trimethylphenanthrene (mean = 1.32 ng m$^{-3}$), and methylated dibenzothiophenes, e.g., methyldibenzothiophene (mean = 1.29 ng m$^{-3}$), dimethyldibenzothiophene (mean = 1.27 ng m$^{-3}$), and trimethyldibenzothiophene (mean = 1.32 ng m$^{-3}$). In dissolved water, methylnapthalene and tetramethylnaphthalene were the most abundant PAHs with average concentrations of 1.12 and 1.45 ng L$^{-1}$, respectively. Measured values of $C_G$, $C_A$, and $C_W$ are known to vary with the changes of each PAH concentration in the marine environment (Berrojalbiz et al., 2011; Castro-Jimenez et al., 2012; Cabrerizo et al., 2014). However, there were no previous reports about the occurrence of PAHs in the Russian Arctic atmosphere and ocean.

The average dry deposition flux ($F_{DD}$) of the $\sum_{35}$ PAHs was 1108 ng m$^{-2}$ d$^{-1}$. The increasing values of $V_d$ may influence $F_{DD}$ in the marine environment due to the higher hydrophobicity of organic compounds, surface microlayer with reduced surface tension, and lipid floating (del Vento and Dachs, 2007b). The higher average $V_d$ was observed for 9-methlyfluorene (1.01–10.02 cm s$^{-1}$), followed by 1,7-dimethylfluorene (1.06–10.63 cm s$^{-1}$) (**Fig. S6**). On a global scale, higher $V_d$ values were found for heavier PAHs such as methylchrysene (0.17–13.30 cm s$^{-1}$) and dibenzo(a,h)anthracene (0.29–1.38 cm s$^{-1}$) (Gonzalez-Gaya et al., 2014). The $V_d$ values reported previously ranged from 0.08 to 0.3 cm s$^{-1}$ in the Atlantic Ocean (Del Vento and Dachs, 2007a) and from 0.01 to 0.8 cm s$^{-1}$ in coastal areas (Holsen and Noll, 1992; Bozlaker et al., 2008; Esen et al., 2008; Eng et al., 2014); the higher values were observed in concentrated industrial and urban areas (Bozlaker et al., 2008). In our study, the highest $V_d$ values were observed in the Barents Sea; the other three seas had similar $V_d$ values ($p > 0.05$) that were lower than in the Barents Sea except for 9-methlyfluorene and 1,7-dimethylfluorene. The East Siberian Sea exhibited the lowest value of $V_d$, while the relatively higher $V_d$ values were found for heavier PAHs (dibenzo(a,h)anthracene, indeno(1,2,3-cd)pyrene, dibenzo(a,h)anthracene and benzo(g,h,i)perylene) in all seas (**Fig. S7**). This may be due to heavier PAHs being principally deposited via heavier aerosols with a higher $V_d$ because they are bound to hydrophobic aerosols or because of gravity, e.g. soot carbon (Gonzalez-Gaya et al., 2014). Dry deposition is a major process for high-molecular-weight PAHs bound to aerosols (**Fig. 6 and Fig. S8**). The deposition values varied mainly in accordance with the PAH concentrations of aerosols in suspension and the factors influencing the deposition velocities (wind speed, compound vapor pressure, etc.). The $F_{WD}$ of the $\sum_{35}$ PAHs ranged from 14 to 19 μg m$^{-2}$ d$^{-1}$. Gonzalez-Gaya et al. (2014) found the highest $F_{WD}$ of $\sum_{64}$ PAHs in the North Atlantic Ocean (24 μg

$m^{-2}$ $d^{-1}$) with an average $F_{WD}$ value of ~8 µg $m^{-2}$ $d^{-1}$ on the global scale based on rain samples. The higher $F_{WD}$ values of PAHs were found in urban areas in China (62.6 µg $m^{-2}$ $d^{-1}$) (Wang et al., 2016) from rain samples and the lower $F_{WD}$ values of PAHs (0.02–0.28 µg $m^{-2}$ $d^{-1}$) from both rain and snow samples were observed in high mountain European areas (Arellano et al., 2018). Our $F_{WD}$ values were within the range of previously reported global scale and the difference in wet deposition was mainly dependent on source distance and precipitation intensity. Wet deposition is an important purging process for semivolatile organic compounds such as PAHs in the gas and aerosol phase. Snow events are quite sporadic in the Arctic Ocean and thus, these have lower relevance for wet deposition of PAHs in this region.

The estimated $F_{AW}$ values revealed that most PAHs had a net input from the atmosphere to the ocean except for the more volatile PAHs, such as 2–3 ring PAHs (**Fig. 7**). The lighter PAHs (2–3 rings) appeared more volatile (978–4892 ng $m^{-2}$ $d^{-1}$) while heavier PAHs (4–6 rings) showed net deposition (1561–7808 ng $m^{-2}$ $d^{-1}$), except dibenzo(a,h)anthracene (1322 ng $m^{-2}$ $d^{-1}$), indeno(1,2,3-cd)pyrene (1238 ng $m^{-2}$ $d^{-1}$), trimethylphenanthrene (1901 ng $m^{-2}$ $d^{-1}$) and benzo(g,h,i)perylene (2708 ng $m^{-2}$ $d^{-1}$). Three orders of magnitude higher net deposition was observed for methylphenanthrene, dimethylphenanthrene, trimethylphenanthrene and tetramethylphenanthrene (**Fig. S9**). Our results were similar to those observed in other PAH-affected areas such as the southeast Mediterranean (Castro-Jimenez et al., 2012), Narragansett Bay (Lohmann et al., 2011) and the North Atlantic Ocean (Lohmann et al., 2009). Ma et al. (2013) suggested that slight volatilization of lighter PAHs may exist from additional sources such as ship ballast and riverine runoff, which is consistent with our findings that volatilization occurred mainly in the East Siberian Sea and Barents Sea, where more industrial factories and urban areas are situated. Our study is also consistent with previous reports showing that diffusion during air–water exchange is the main process for transfer of relatively light volatile organic compounds to the marine environment (Castro-Jimenez et al., 2012; Jurado et al., 2005). The integrated monthly $F_{AW}$ (tons per month) of 5–6 ring PAHs were comparable to $F_{DD}$ values in the Russian Arctic Ocean, whereas only the East Siberian Sea showed high levels of dry deposition (**Fig. S9**). In all four seas, $F_{AW}$ values of 3–4 ring PAHs were of the same magnitude as $F_{DD}$ values. The total volatilization and total adsorption of the $\sum_{35}$ PAHs in the Russian Arctic Ocean amounted to 2600 tons per month and 3640 tons per month, respectively. Therefore, there was a net input of the $\sum_{35}$ PAHs from the atmosphere to the marine environment that reached 3276 tons, which was 100 times higher than for the aerosol-bound $\sum_{35}$ PAHs that underwent dry deposition (estimated at ~30 tons per month). In other reports, Gonzalez-Gaya et al. (2016) estimated the global input of PAHs from the atmosphere to the ocean to be on the order of 90,000 tons per month and Reddy et al. (2012) reported that the input of PAHs to the ocean in the Gulf of Mexico in 2010 after the Deepwater Horizon oil spill was 20,000 tons. Such comparisons suggest that the diffusive fluxes in the Russian

Arctic Ocean play an important role in the atmosphere–ocean exchange of PAHs, whereas there is a lower input of PAHs to the Russian Arctic Ocean on the global scale.

In addition to the transfers of PAHs to the ocean, PAHs also can be degraded during transport through the atmosphere because of reactions with OH radicals (Keyte et al., 2013). The degradation flux $D_{atm}$ of PAHs in the gas phase of the oceanic atmosphere (**Fig. 8**) was estimated at 3000 tons per month for the $\sum_{35}$ PAHs, which represents an additional PAH sink (see Methods). In general, the large amounts of PAHs undergoing net deposition to the ocean and degradation during atmospheric transport are indicative of large source areas in the Russian Arctic. Notably, PAHs concentrations have been increasing in the atmosphere owing to wildfires and fossil fuel use over the past century (Zhang and Tao, 2009). The high-molecular weight PAHs were dominant in the aerosols (**Fig. 5**) originating from pyrolytic sources (Lima et al., 2005). Besides, high abundances of alkylated PAHs were observed in the gas and dissolved phases, and along with the evaluations of the diagnostic ratios (**Fig. S10**), the results were suggestive of pyrogenic sources for PAHs in gases and aerosols, and mixtures of pyrogenic and petrogenic sources for PAHs in dissolved water (mostly for the Leptev Sea and East Siberian Sea). Other sources contributing to the occurrence of PAHs may have involved both anthropogenic and biogenic sources on land (Cabrerizo et al., 2011). In the Russian Arctic Ocean, it can be assumed that PAHs in the atmosphere (gas and aerosol) originated from anthropogenic sources including industrial and urban activities, while PAHs in seawater, at sites with relatively less anthropogenic sources, i.e., the Leptev Sea and East Siberian Sea, originated from a mixture of anthropogenic and biogenic sources. This indicates that atmospheric transport of PAHs derived from anthropogenic activities occurs in all sectors of the Russian Arctic Ocean while only the East Siberian Sea and Leptev Sea have more anthropogenic PAHs in the water phase.

Because PAHs are toxic, these chemicals can have an adverse influence on food webs in marine ecosystems (Hylland, 2006). In particular, even though PAHs are present at natural background levels in the marine environment, the massive usage of fossil fuels has led to increases in PAH emissions and excessive PAH concentrations in many marine environments. The present study indicates that there are high contributions of diffusive atmospheric PAHs to the Arctic Ocean, and these chemicals are potentially perturbing the carbon cycle in the ocean and posing risks to the fragile Arctic marine food webs. Thus, further studies of the impacts of such chemicals are warranted.

## 4 Conclusion

This study presents the occurrence and atmosphere–ocean fluxes of 35 PAHs and 9 heavy metals in the Arctic Ocean. Dry deposition and wet deposition fluxes of 9 heavy metals in aerosols were estimated at 2205 ng m$^{-2}$ d$^{-1}$

and 10.95 μg m$^{-2}$ d$^{-1}$, respectively. The net gross absorption of Hg in the Arctic Ocean was estimated at 280 tons per month. A net input of PAHs from the atmosphere to the Arctic Ocean was observed for most of the PAHs, especially for the low-molecular weight PAHs. The net atmospheric input of the 35 PAHs was estimated at 3276 tons per month. The current occurrences of semivolatile aromatic hydrocarbons could have been derived from biogenic sources and anthropogenic sources from continental land masses, especially for the locations close to industrial areas. These inputs of Hg and PAHs may be causing adverse effects on the fragile Arctic marine ecosystems; this issue warrants further research. In addition, our study suggests that both PAHs and metals are affected by local depositional effects and change of source emission, and therefore, spatial distribution of these compounds and source identification need to be further investigated.

**Author contributions.** Dr. Xiaowen Ji and Dr. Evgeny Abakumov set up the sampling equipment and analyzed the samples, and the data. Dr. Xianchuan Xie also helped to collect and analyze the data. Dr. Xiaowen Ji and Dr. Xianchuan Xie wrote this manuscript.

**Competing interests.** The authors declare that they have no conflict of interest.

**Acknowledgements.** This work was supported by grants from the Russian Foundation for Basic Research (18-44-890003, 16-34-60010); by a grant from Saint-Petersburg State University titled "Urbanized ecosystems of the Russian Arctic: dynamics, state and sustainable development"; by the Jiangsu Nature Science Fund (BK20151378), and by the Fundamental Research Funds for the Central Universities (090514380001). We would like to thank Miss Yu Su from the School of Visual Art, BFA Computer Art for helping to draw supplementary journal cover, and Miss Kuznetsova Ekaterina for helping with the Russian translation.

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

**Figure caption list**

**Figure 1**. Locations of investigated islands for soil sampling and trajectory of the vessel in the Russian Arctic.

**Figure 2.** Occurrence of heavy metals. Results show the concentrations of heavy metals in the **(a)** gas phase, **(b)** aerosol phase, **(c)** and dissolved water phase. Colored bars show the sum of nine quantified metals. The number at the bottom of the legend bars represent the concentration scale as same as Figure 3-7.

**Figure 3.** Measured atmosphere–ocean exchange of heavy metals. **(a)** Fluxes of dry deposition for nine heavy metals; **(b)** fluxes of net diffusive air–water exchange for Hg. In panel **(a)**, colored bars represent the sum of nine heavy metals. In panel **(b)**, downward bars represent the net deposition into the ocean, and upward bars represent the net volatilization of Hg.

**Figure 4.** Wet deposition of **(a)** heavy metals and **(b)** PAH fluxes. The measured wet deposition of heavy metals and PAHs occurred during the eight snow events encountered during the vessel expedition.

**Figure 5.** Occurrence of PAHs in the Russian Arctic Ocean. Concentrations of PAHs in the **(a)** gas phase, **(b)** aerosol phase, and **(c)** dissolved water phase. Color bars indicate the sum of 35 PAHs, where each PAH corresponds to the bottom legend (colors range from red for the heaviest molecular weight PAHs to green for the lightest molecular weight PAHs).

**Figure 6.** Dry deposition fluxes for the 35 measured PAHs. Color bars indicate the sum of the 35 quantified compounds, and each color represents the individual PAHs in the bottom legend (colors range from red for the heaviest molecular weight PAHs to green for the lightest molecular weight PAHs).

**Figure 7.** Measured atmosphere–ocean exchange of PAHs. **(a)** net diffusive air–water exchange fluxes (all net deposition into the ocean) and **(b)** volatilization air–water exchange fluxes. Color bars indicate the sum of the 35 quantified compounds, and each color represents the individual PAHs in the bottom legend (colors range from red for the heaviest molecular weight PAHs to green for the lightest molecular weight PAHs).

**Figure 8.** Atmospheric degradation of PAHs. Estimated fluxes of degraded PAHs in the gas phase following reaction with OH radicals.

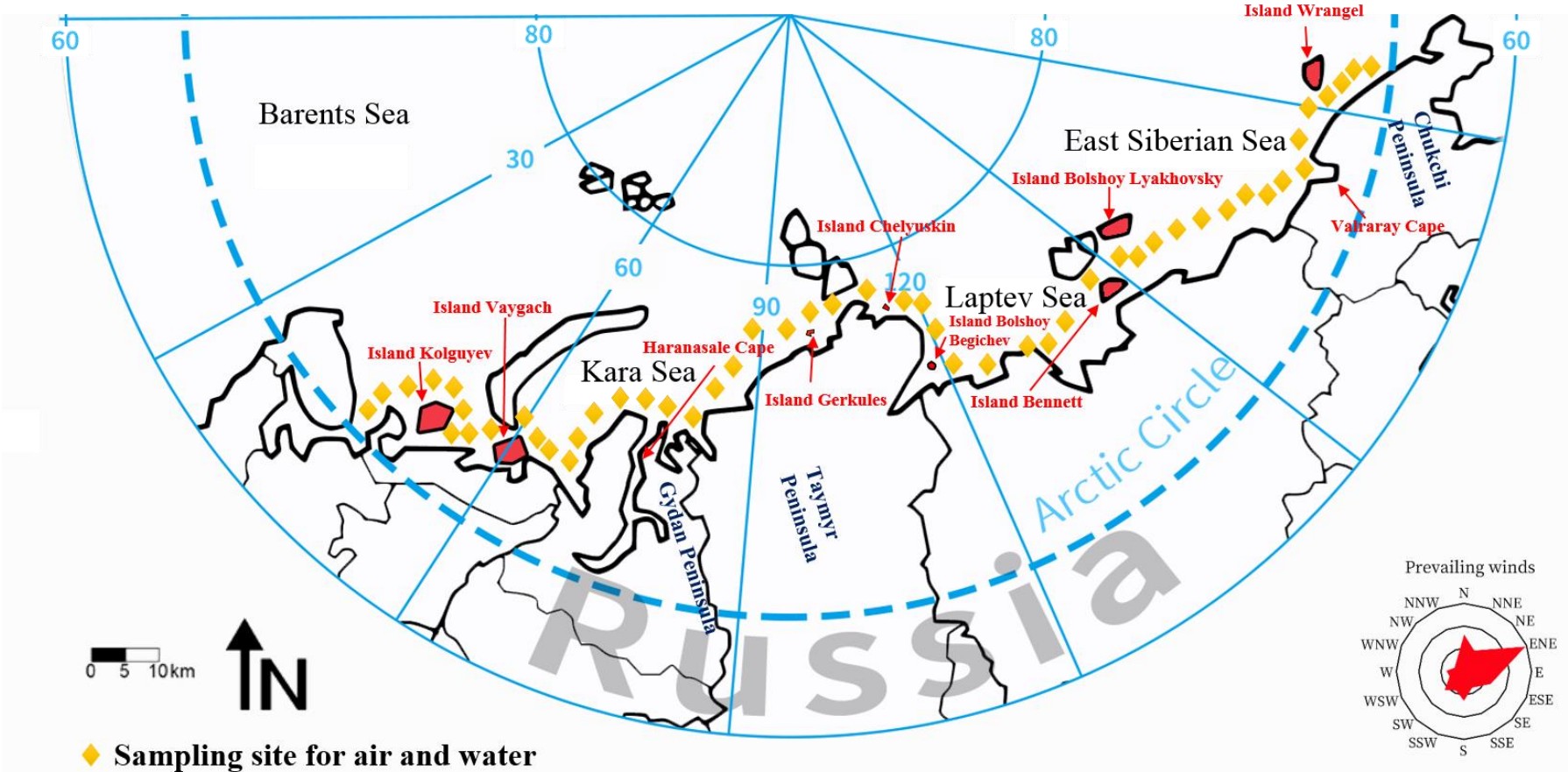

**Figure 1.**

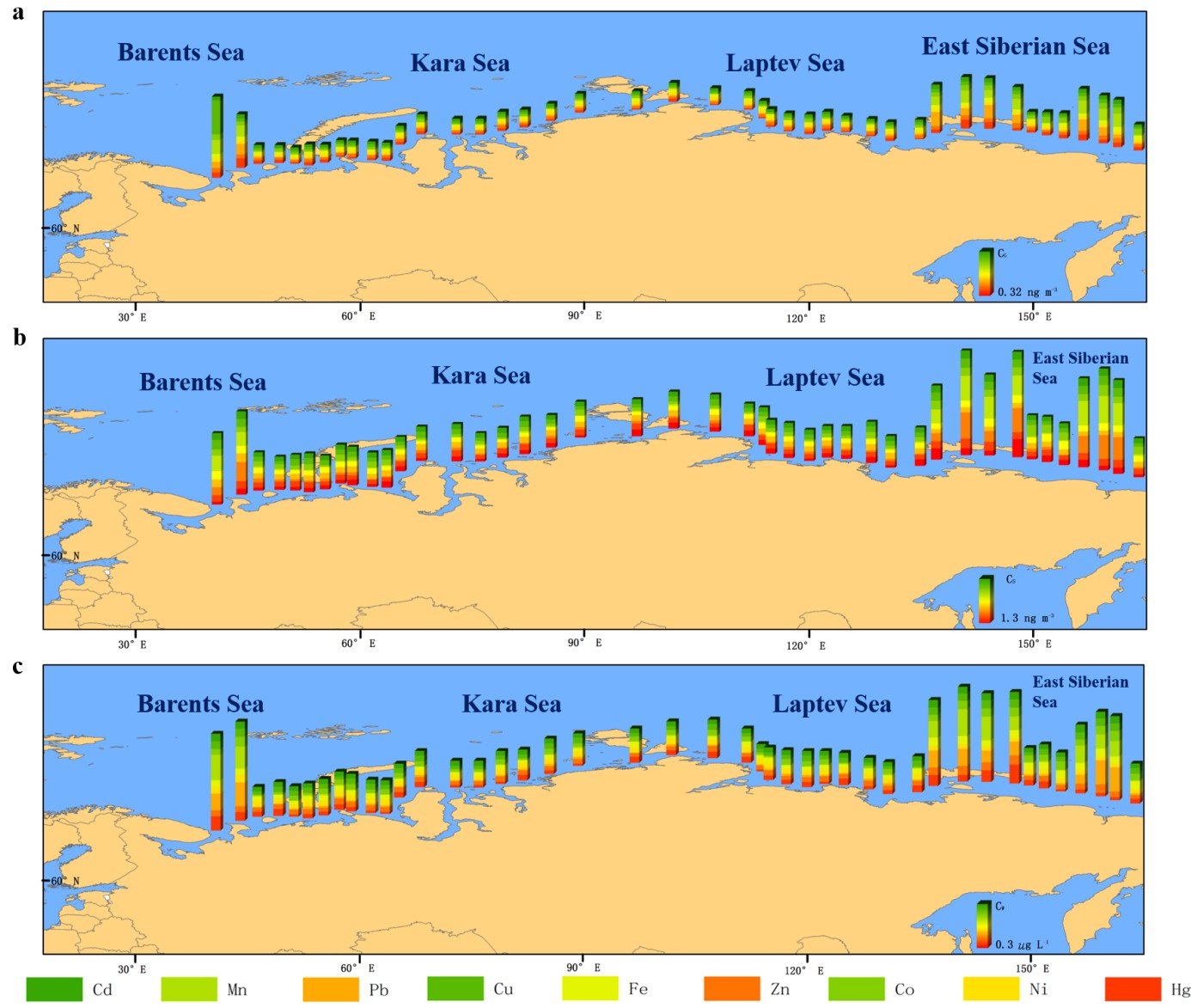

**Figure 2.**

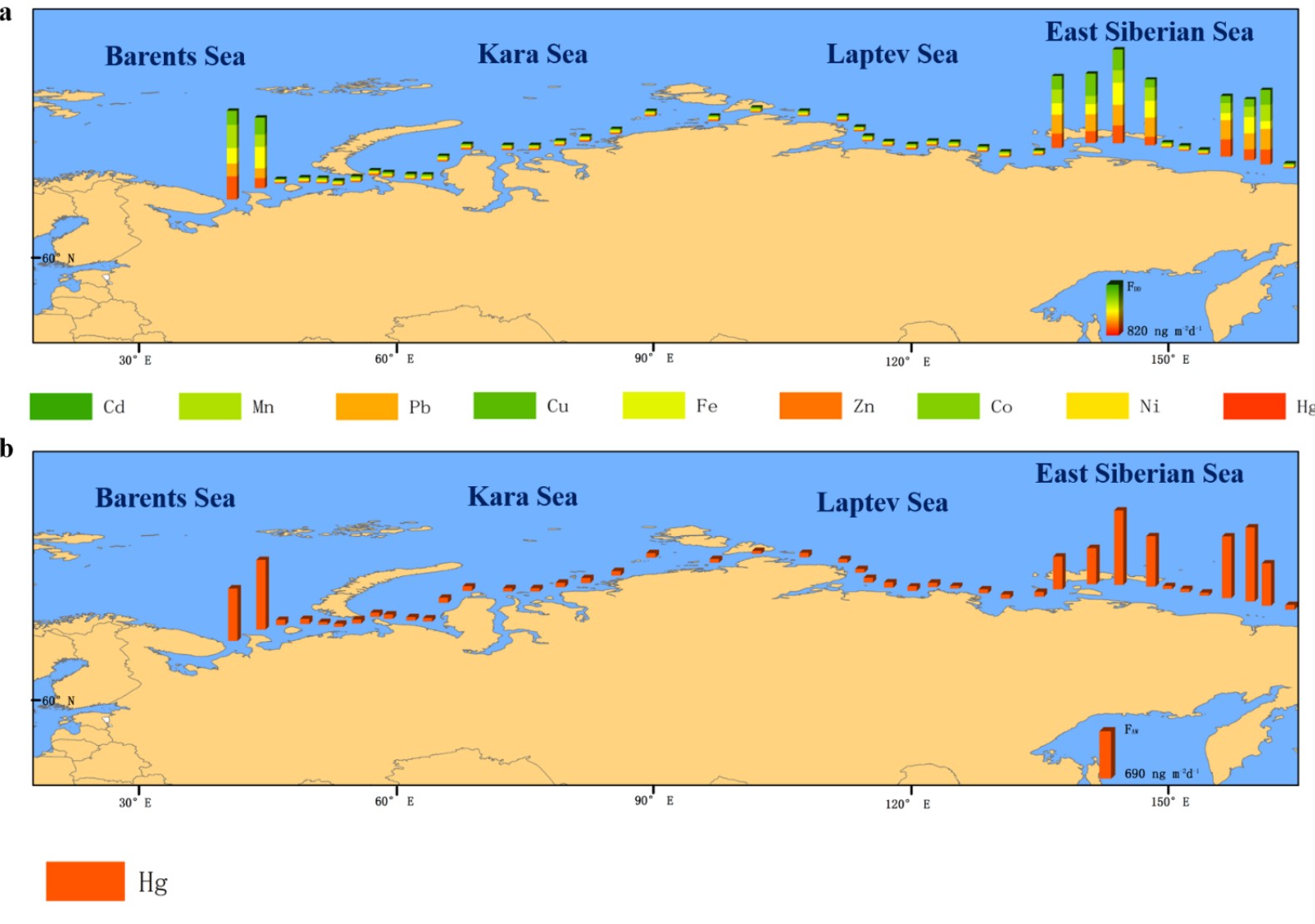

**Figure 3.**

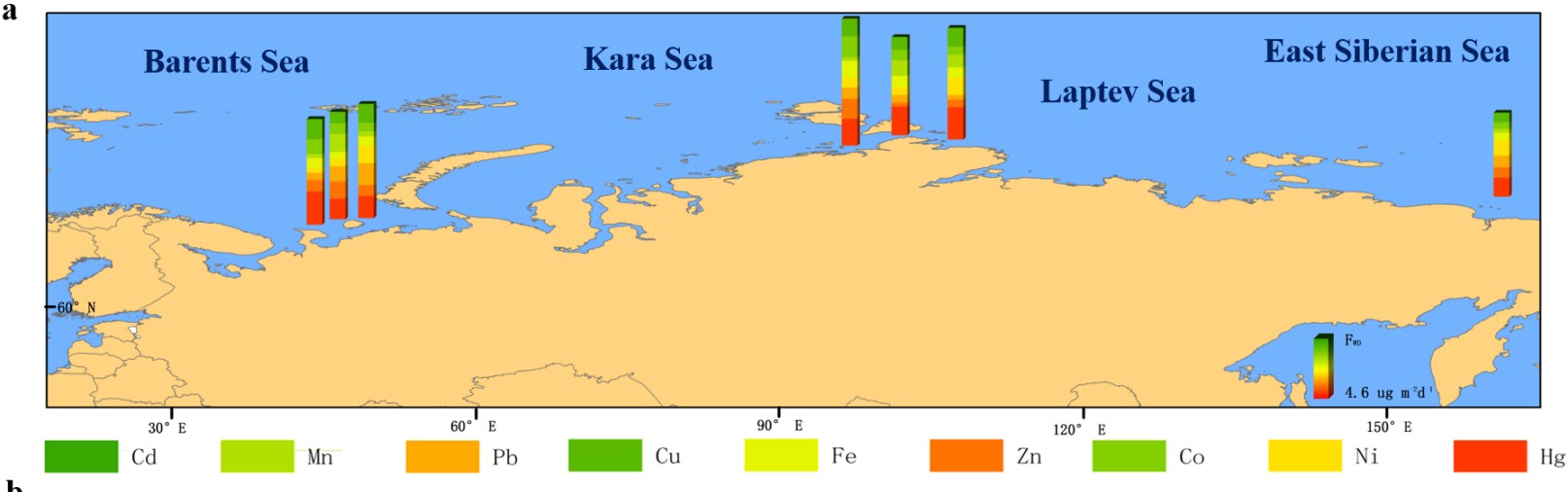

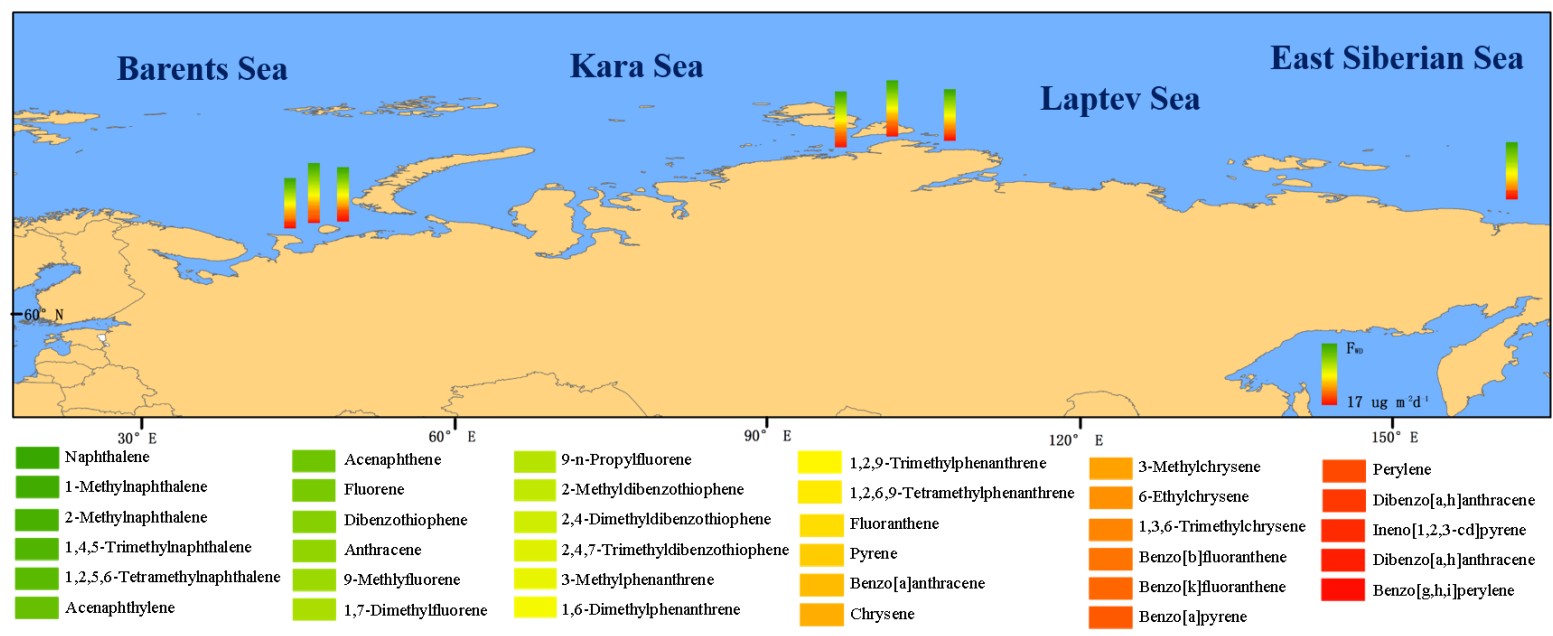

**Figure 4.**

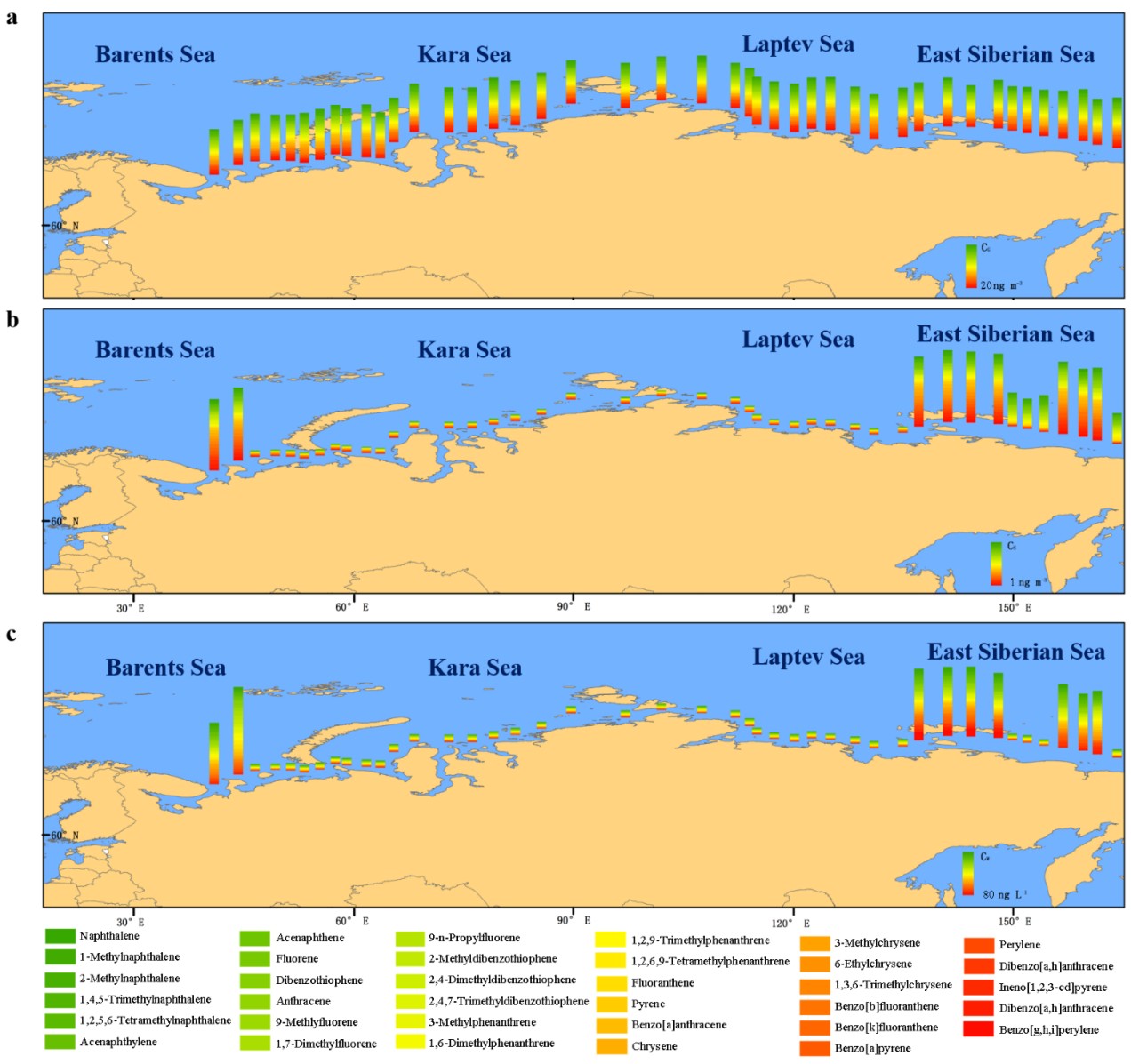

**Figure 5.**

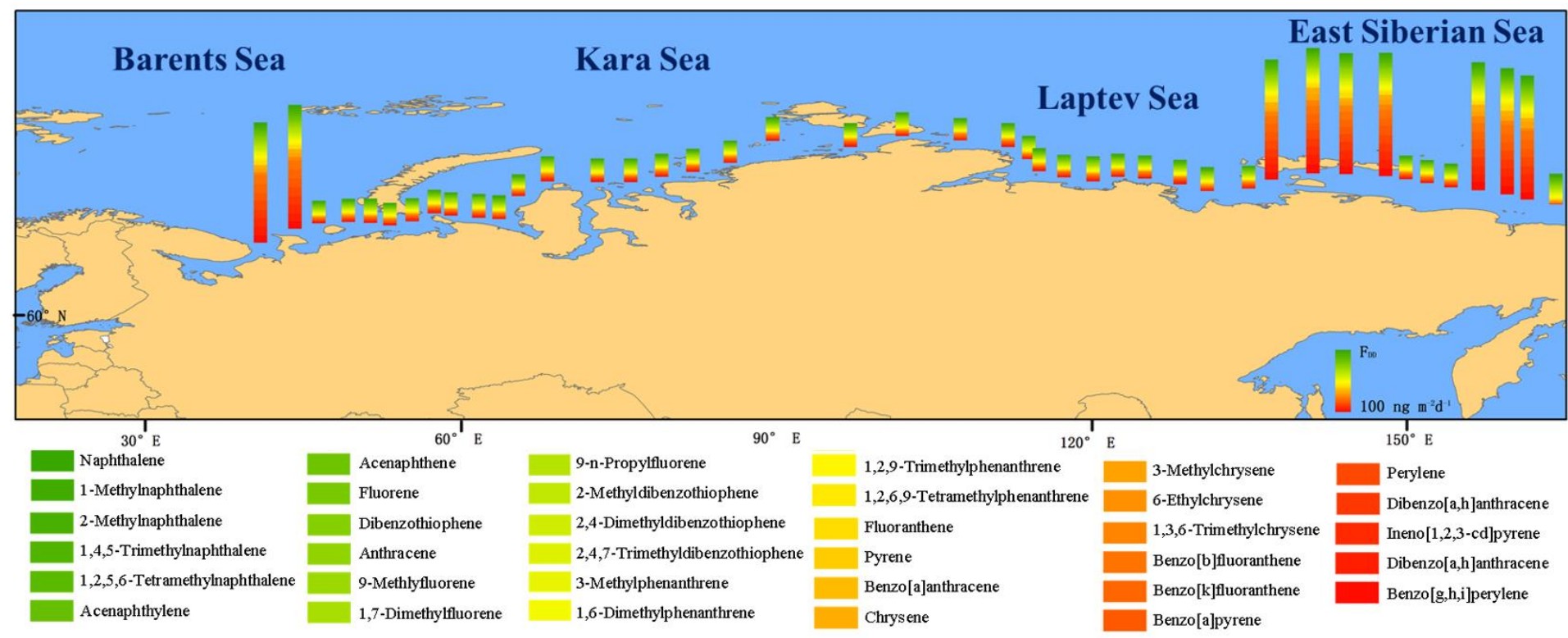

**Figure 6.**

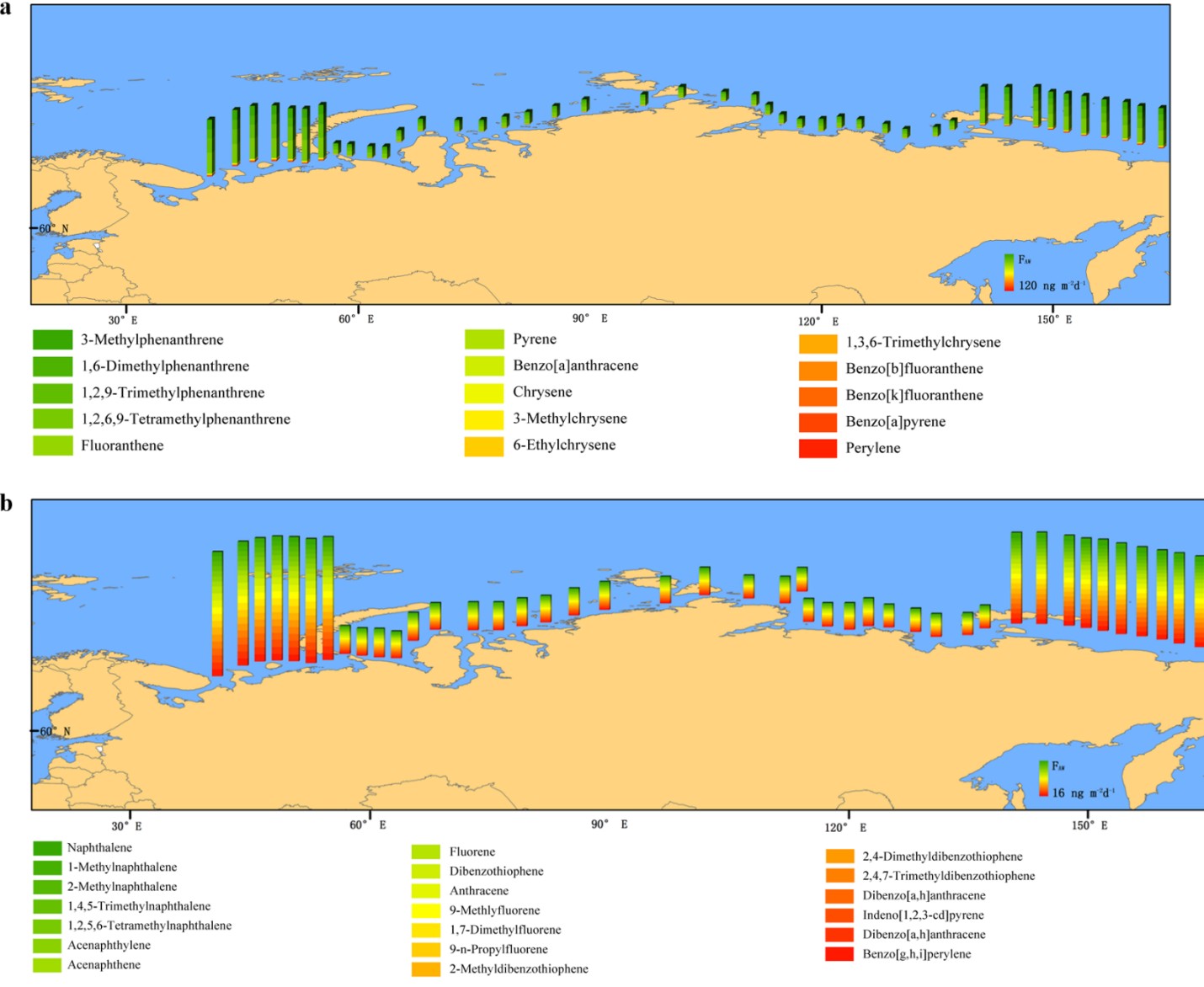

**a**

3-Methylphenanthrene
1,6-Dimethylphenanthrene
1,2,9-Trimethylphenanthrene
1,2,6,9-Tetramethylphenanthrene
Fluoranthene

Pyrene
Benzo[a]anthracene
Chrysene
3-Methylchrysene
6-Ethylchrysene

1,3,6-Trimethylchrysene
Benzo[b]fluoranthene
Benzo[k]fluoranthene
Benzo[a]pyrene
Perylene

**b**

Naphthalene
1-Methylnaphthalene
2-Methylnaphthalene
1,4,5-Trimethylnaphthalene
1,2,5,6-Tetramethylnaphthalene
Acenaphthylene
Acenaphthene

Fluorene
Dibenzothiophene
Anthracene
9-Methlyfluorene
1,7-Dimethylfluorene
9-n-Propylfluorene
2-Methyldibenzothiophene

2,4-Dimethyldibenzothiophene
2,4,7-Trimethyldibenzothiophene
Dibenzo[a,h]anthracene
Indeno[1,2,3-cd]pyrene
Dibenzo[a,h]anthracene
Benzo[g,h,i]perylene

**Figure 7.**

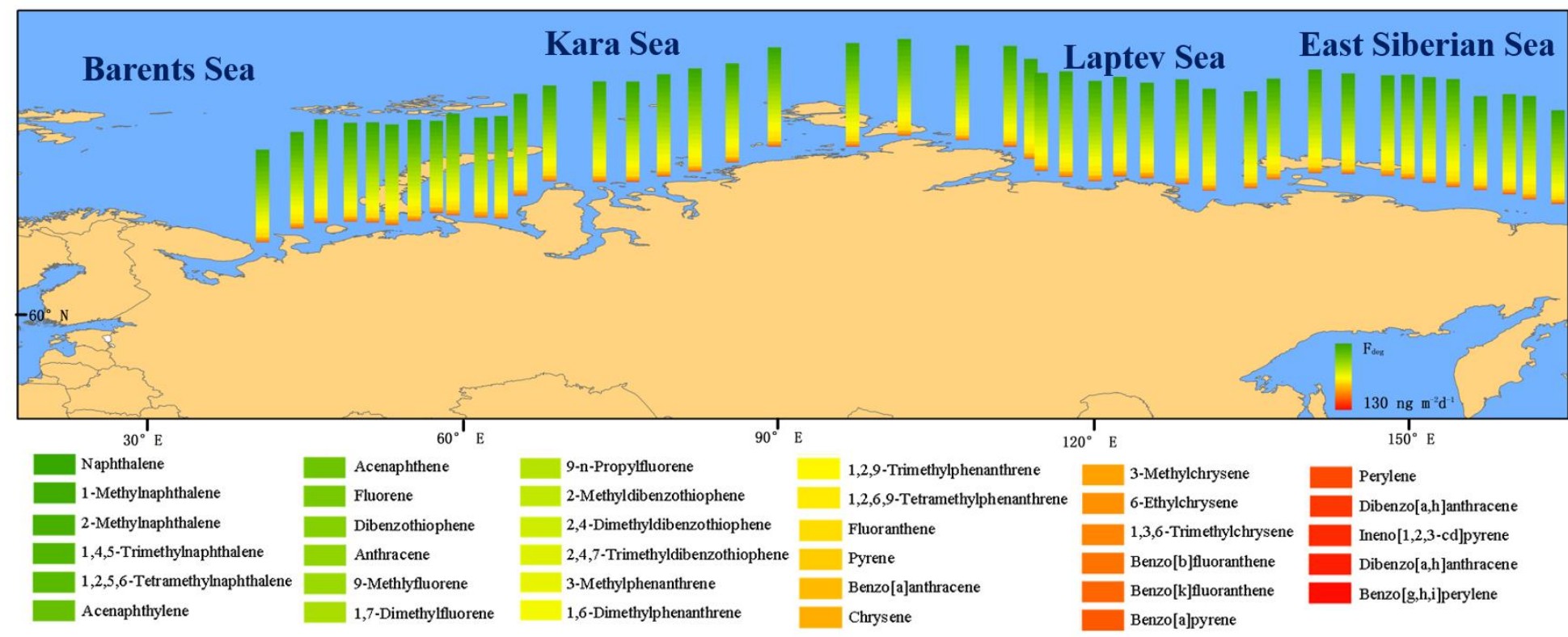

**Figure 8.**