# Peer review of "hydrocarbons in the Russian Arctic Ocean"

_Atmospheric Chemistry and Physics, 2019_

## Referee Comment (RC1) · Anonymous Referee #1 · 14 Aug 2019

The MS titled "Atmosphere–ocean exchange of heavy metals and polycyclic aromatic hydrocarbons in the Russian Arctic Ocean" written by Ji et al. researched the results of a Russian Arctic assessment of the occurrences and atmosphere–ocean fluxes of 35 polycyclic aromatic hydrocarbons (PAHs) and 9 metals (Pb, Cd, Cu, Zn, Fe, Mn, Ni, and Hg). The topic of this study is quite interesting due to the reflection of anthropogenic influences in the normal biochemical cycles as the balance of geochemical substances between ocean and gas. Authors pointed out the net input of Hg and 35 PAHs into ocean, filling the data gaps in this field. The deposition including dry and wet deposition in the Arctic Ocean so far appeared very sporadic data without a continuous sampling sites and period. The topic of this MS is within scope of ACP, the language usage and

structure of article is generally good and well-written. However, there are still some minor problems and some questions remained. I would like to support this MS to be published unless my questions and problems are well addressed or answered. There are specific questions I found in this MS.

L43-44, it's quite not logic here, authors should introduce the transported pollutants from low attitude to the polar region as the Arctic rather than the present statements about reducing global emissions. Additionally, global emissions of the atmospheric pollutants were carried by monsoon to the high latitude. What is the contribution of it to the Arctic area? Are there any reports concerning the separate contribution of airborne pollutants to land and ocean? L46, the reason mercury is key problematic pollutant should be briefly mentioned here. And citation is needed here. L47, what kind of sources are referring here? L49, "the melting of contaminated ice" in the ocean or also in the terrestrial land? L68, what is relevant connection between benthic input of metals and air-seawater exchange input? L88-89, the sentence authors make here tried to indicate the inorganic salt ions may increase during the summer melting season, and thence the organic compounds' solubility would change? Now the meaning is not clear. Authors need to rephase the sentences to clarify. L91, need citation for "the Arctic Ocean is considered as a sink that receives global airborne pollutants". L203, how surface chlorophyll concentrations were calculated or measured? L218, why H' can be corrected by the salinity? In Fick's law, I am not aware of such connection. H' values are usually considered by the temperature changes. L246, how the uncertainty of air-water exchange net direction was conducted? L263, why only OH radicals were considered to the degradation of PAHs in the atmosphere? L279-280, Do author have any proofs to support the reason with air trajectories of Russian Arctic? L294, which sea Taymyr Peninsula is closer to? Clarify here. L295, I think Shevchenko et al. 2013 should be moved to previous sentence. L419, Can authors explain what molecular weight PAHs were easier to de degraded in the air because it's important for biocycles to consume these carbon sources by organisms. If most of heavy molecular PAHs enter ocean ecosystem, which may not be consumed by biota. L440, I suggest authors to

put some information of future protective measures for this region.

---

## Short Comment (SC1) · 27 Aug 2019

Dear Authors,

Your paper titled "Atmosphere–ocean exchange of heavy metals and polycyclic aromatic hydrocarbons in the Russian Arctic Ocean" is interesting. I believe the data can be fill the gap of net input/output of these substances originating from both anthropogenic sources and natural sources. The clarification and structure of the paper is clear and generally good. I have some questions concerning to this paper.

1. How far the samples were collected between two sample sites?

2. How authors choose these sites?

3. Authors mentioned the atmospheric mercury depletion events (AMDEs) of Hg in the paper, so were there any sites collected during the sunrise in spring season?

4. Do authors try to avoid this time?

5. Were Dissolve Organic Carbon(DOC) and the concentration of surface chlorophyll same sample as the water?

6. How authors can make sure the corresponding the water sample to air samples?

7. Authors calculated the atmospheric oxidation of PAHs initiated by reaction with the OH radical in gas phase?

8. Is it possible to be also degradable in the particle phase?

Best regards,

Chunlan Li

---

## Short Comment (SC2) · 9 Sep 2019

Dear authors, it's an interesting report about PAHs and metals net deposition in the Arctic Ocean. I would like to draw your attention to some deficiencies. Firstly, PAHs part seems to get some similar conclusion with the one (Belén González-Gaya et al. 2016, High atmosphere–ocean exchange of semivolatile aromatic hydrocarbons, nature geoscience) while that one was established in the global scale. This study can support their results to fill the gap in Arctic Ocean. While the metals' part needs a major revision for discussion with other ocean and metals change between all detected ones. So far, it only seems to focus on mercury. There are some paper authors should

study, as below (and others):

Tuohy, A., Bertler, N., Neff, P., Edwards, R., Emanuelsson, D., Beers, T., and Mayewski, P. ( 2015), Transport and deposition of heavy metals in the Ross Sea Region, Antarctica, J. Geophys. Res. Atmos., 120, 10,996– 11,011, doi:10.1002/2015JD023293.

Jurina, I.; Ivanić, M.; Vdović, N.; Troskot-Čorbić, T.; Lojen, S.; Mikac, N.; Sondi, I. (2015), Deposition of trace metals in sediments of the deltaic plain and adjacent coastal area (the Neretva River, Adriatic Sea). Journal of Geochemical Exploration. DOI: 10.1016/j.gexplo.2015.06.005.

Authors need to revise Figure 6 b, it's net deposition from air to sea, the bar should be downwards.
* * *

---

## Referee Comment (RC2) · Anonymous Referee #2 · 10 Sep 2019

Review of "Atmosphere-ocean Exchange of heavy metals and polycyclic aromatic hydrocarbon in the Russian Arctic Ocean".

This work reports the concentrations of PAHs and metals in air, water and snow in the Russian sector of the Arctic. There is no previous data for this sector, so this contribution is very important. The work is of mix quality, with sections that are generally well done, but other material that is erroneous or needs further work. The manuscript needs some work in order to present the data appropriately, improve the comparison with other studies for polar regions, and give some extra depth to the discussion. This revision is mainly for the PAHs part. I suggest moderate to major modifications before

it can be accepted.

- Line 40. Comment and cite a work on long-range atmospheric transport of PAHs (or pops) to the arctic.

- Line 49-50. Provide examples, there are some published for Antarctica and the Arctic for both metals and pahs.

- Note that not all PAHs are persistent, it also depends if they are found in the gas or dissolved phase (less persistent) or associated to aerosols and particulate matter.

- Line 73. Biogeochemical instead of biochemical

- Line 77-79. Specify if you are referring to rain or snow, or both. I guess snow.

- Line 110. This citation is not adequate here.

- Line 116. I guess that first in aluminum foil and after in polyethylene bags, which must be air-tight.

- Line 119. Was snow melted immediately? How was snow melted?

- Line 123. M3 or L, these are huge volumes. I have never seen such large volumes for water using a XAD. Unless there is a typo mistake, such volumes need a justification and discussion.

- Line 265. It cannot be equation 13.

- Line 266. Is this concentration appropriate for the arctic? Discuss.

- Line 269-270. I cannot understand this sentence if these estimations have just been explained.

- Line 370. In addition to the highest, provide the range for each basin. Generally, the values commented here do not correspond always to the values seen in Figure 5.

- Line 372. The concentrations in aerosols are extremely high! These need a compar-
ison with other studies and a discussion. Which is the black carbon concentration in aerosol for this region?

- Generally, compare the pahs and metal concentrations with other reports for the Arctic, even if these are for the north atlantic and north pacific.

- Line 391. Here or in the methods, comment the range of estimated dry deposition velocities.

- Line 395. Huge values, If possible compare with other measures. Report as well the concentrations in snow.

- Line 398. Comment the range of diffusive fluxes, and show them in a figure, maybe in the supplementary material.

- Line 399 and figure 6b. This is not clear to me. For which pahs there is a net volatilization and for which there is a net deposition.

- Line 413. Rewrite. As I understand them, these fluxes for the the basins studied, but not all the Arctic, which should be clarified. Provide the surface for each basin in methods.

- Review the spelling of pah's names in figure 4 and 5. The size of the legend (scale) in the figures should be bigger.

- Review the use of English in the manuscript.

- Table S4. Why dibenzothiphene and anthracene appear twice in this table, while phenanthrene is not there? Review all the tables and data set. In addition, the mean total concentrations seem to not correspond to the distribution seen in Figure 5.

- Figure S5. The three legends appear as Cg. . . correct.

- Improve the quality of the figures in the supplementary material. Generally, the presentation aspects of this work need to be reviewed.

---

## Author Comment (AC2) · 24 Sep 2019

Review of "Atmosphere-ocean Exchange of heavy metals and polycyclic aromatic hydrocarbon in the Russian Arctic Ocean".

This work reports the concentrations of PAHs and metals in air, water and snow in the Russian sector of the Arctic. There is no previous data for this sector, so this contribution is very important. The work is of mix quality, with sections that are generally well done, but other material that is erroneous or needs further work. The manuscript needs some work in order to present the data appropriately, improve the comparison with other studies for polar regions, and give some extra depth to the discussion. This revision is mainly for the PAHs part. I suggest moderate to major modifications before it can be accepted.

**Response**: we are very grateful that you spend your valuable time helping us to revise this manuscript with good comments and suggestions.

1. Line 40. Comment and cite a work on long-range atmospheric transport of PAHs (or pops) to the arctic.

**Response**: We will add the references concerning long-range atmospheric transport of PAHs (or POPs) to the arctic.

2. Line 49-50. Provide examples, there are some published for Antarctica and the Arctic for both metals and pahs.

**Response**: We will add the published paper for metals and PAHs in polar regions as examples.

3. Note that not all PAHs are persistent, it also depends if they are found in the gas or dissolved phase (less persistent) or associated to aerosols and particulate matter.

**Response**: thank you for pointing out this important note. In revised manuscript, we will add the examples which clarified that aerosols contained PAHs deposited into sea ice and soils (or snow) which is an important source for the contribution of persistent PAHs. Indeed, according to the previous study (**Fernandes and Sicre, 1999**), the major PAHs either from adjacent areas or low latitudes are in the form of aerosols and binding with black carbon (particulate matter).

Fernandes, M. B., and Sicre, M. A.: Polycyclic Aromatic Hydrocarbons in the Arctic: Ob and Yenisei Estuaries and Kara Sea Shelf, Estuarine, Coastal and Shelf Science, 48, 725-737, https://doi.org/10.1006/ecss.1999.0472, 1999.

4. Line 73. Biogeochemical instead of biochemical

**Response**: we will revise it.

5. Line 77-79. Specify if you are referring to rain or snow, or both. I guess snow.

**Response**: we will specify the wet deposition as snow deposition in **re**vised manuscript.

6. Line 110. This citation is not adequate here.

**Response**: we realized the methods here concerning simultaneous sampling both for gas and aerosol phase PAHs. The citations of details about these two methods we will add.

7. Line 116. I guess that first in aluminum foil and after in polyethylene bags, which must be air-tight.

**Response**: after air sampling in the ship, we did keep the samples in folded aluminium foil firstly. We will revise this sentence as "the filters and PUFs were firstly covered with aluminum foil tightly for air-tightness, then immediately placed in polyethylene bags and zip the bags, and frozen at -20 °C prior to chemical analyses."

8. Line 119. Was snow melted immediately? How was snow melted?

**Response**: no, snowfall samples were not melted immediately. After sampling, all samples were protected from light and stored at 4 °C in borosilicate glass bottles prior to the analysis. Snowfall samples were melted thoroughly at room temperature. We will add this information

9. Line 123. M3 or L, these are huge volumes. I have never seen such large volumes for water using a XAD. Unless there is a typo mistake, such volumes need a justification and discussion.

**Response**: Thank you for pointing out this mistake. The unit should be "mL". This mistake will be corrected. We only used XAD-2 resin rather than XAD-4 resin for high volume of water. Water was filtered at a flow rate averaging 4 L/min with a pneumatic pump (Flojet) through a borosilicate microfiber glass filters (1 mm nominal pore size), housed in an aluminum filter support.

10. Line 265. It cannot be equation 13.

**Response**: we are sorry for this mistake. The hydroxyl radicals concentrations [OH] in the considered mixed layer (between 1,000 and 500 hPa) was based on the zonally and monthly averaged concentrations of OH radicals from **Spivakovsky and coworkers (2000)**. Therefore, this sentence has been removed.

Spivakovsky, C. M. et al. Three-dimensional climatological distribution of tropospheric OH: update and evaluation. J. Geophys. Res. 105, 8931-8980 (2000).

11. Line 266. Is this concentration appropriate for the arctic? Discuss.

**Response**: This is a very good question to discuss! We must admit that there is no ranges or exact data of OH radical levels in troposphere of polar regions.

Firstly, the primary source of HO radical in the troposphere is the photolysis of $O_3$ to produce $O$ ($^1D$). This reaction requires radiation of wavelength less than 315nm, otherwise ($O^3P$) is produced. The latter species reacts with molecular oxygen in the presence of $N_2$ or $O_2$ to produce $O_3$. The electronically excited $O(^1D)$ usually relaxes to produce ($O^3P$) but also undergoes reaction with water to produce HO. While minor sources of HO are available through reaction of $O(^1D)$ with $CH_4$ and $N_2$. Therefore, the methods to measure OH concentrations were usually based on observed distributions of $O_3$, $H_2O$, $NO_t$ ($NO_2+NO+2N_2O_5+NO_3 +HNO_2+HNO_4$), CO, hydrocarbons, temperature, and cloud optical depth.

Thus, it is difficult to compare the OH concentrations by different methods and altitudes as well as latitudes. **Hewitt and Harrison (1985)** summarized the OH concentrations in different

altitude as well as the global mean, which showed the range of 0.5-5 × 10$^6$ mol cm$^{-3}$ for daytime OH radical. **Li et al. (2008)** used an empirical method is presented to determine effective OH concentrations in the troposphere and lower stratosphere, based on CH$_4$, CH$_3$Cl, and SF$_6$ data from aircraft measurements (IAGOS-CARIBIC) and a ground-based station (NOAA). The results showed tropospheric OH average values of 10.9 × 10$^5$ (σ = 9.6 × 105) mol cm$^{-3}$ in a global level. The reason we used data from **Spivakovsky and coworkers (2000)** is that the estimation of OH concentrations considered by latitudes (±32$^o$) in Northern Hemisphere.

It should be noted that precipitation amounts in the Arctic are very low (annual precipitation on Svalbard is 150-300 mm). This means that wet scavenging is not efficient and the lifetime for soluble species like aerosols is longer than on the continents to the south. Gas- phase chemical removal of trace compounds all but stops in the Arctic atmosphere during the polar night. In the absence of sunshine, the production rate of the OH radical, which is the main gas-phase scavenger, is low. Also, in the sunlit part of the year, the chemical lifetime of trace compounds is relatively long in the polar atmosphere (except in the near-surface layer that is impacted by snowpack photochemical processes), due to the strong attenuation of short wave visible sunlight as the solar elevation is low and the low specific humidity. Therefore, the OH radicals in the Arctic troposphere should be further measured throughout the year. So far, OH radicals levels are within the global levels.

C.N. Hewitt, Roy M. Harrison, Tropospheric concentrations of the hydroxyl radical—a review, Atmospheric Environment (1967), Volume 19, Issue 4, 1985, Pages 545-554.

Li, Mengze; Karu, Einar; Brenninkmeijer, Carl; Fischer, Horst; Lelieveld, Jos; Williams, Jonathan; 2018; npj Climate and Atmospheric Science, (1): 29.

Spivakovsky, C. M. et al. Three-dimensional climatological distribution of tropospheric OH: update and evaluation. J. Geophys. Res. 105, 8931-8980 (2000).

12. Line 269-270. I cannot understand this sentence if these estimations have just been explained.
**Response**: the uncertainty (error propagation) analysis basically used the relative standard deviation (RSD) based on measured uncertainties in air and water analysis, air−water partitioning coefficients (including Henry's law constant and temperature), and overall mass transfer velocity were considered. We will add the equation in Supplementary Materials.

13. Line 370. In addition to the highest, provide the range for each basin. Generally, the values commented here do not correspond always to the values seen in Figure 5.
**Response**: the range of $\sum_{35}$ PAH concentrations in gas, aerosols and dissolve water for each basin will be added. We have checked the original data in Figure 5, some wrong concentration values will be revised in this paragraph.

14. Line 372. The concentrations in aerosols are extremely high! These need a comparison with other studies and a discussion. Which is the black carbon concentration in aerosol for this region?

**Response**: we are very sorry that this concentration was showing wrong in the paper. It cannot be so high. All concentrations should be divided 100. We will make corrections for graphs and texts. There are some reports concerning PAH concentrations in atmosphere in Ocean. However, not all separating gas and particle phase. Therefore, we only compared and discussed with the separated ones as "$\sum_{35}$ PAH concentrations in aerosols ($C_A$, ng m$^{-3}$) in the Barents Sea (0.25-2.95), and East Siberian Sea (0.24-3.32) with average $C_A$ values of 1.38 and 2.07 ng m$^{-3}$ respectively were apparently higher than those in the Leptev Sea (0.23-0.89) and Kara Sea (0.23-0.27) with average CA values of 0.30 and 0.25 ng m$^{-3}$, respectively (Fig. 5b). The average CA of $\sum_{35}$ PAH in present study is higher than those of $\sum_{64}$ PAH measured in South Atlantic Ocean (mean = 0.93 ng m$^{-3}$) and North Pacific Ocean (mean = 0.56 ng m-3) while much lower than those of $\sum_{64}$ PAH in Indian Ocean (mean = 10 ng m-3) (Gonzalez-Gaya et al., 2016). Considering average $\sum_{35}$ PAH $C_A$ (1.02 ng m$^{-3}$) in the Russian Arctic Ocean, the value is comparable to those of $\sum_{64}$ PAH observed in South Atlantic Ocean and South Pacific Ocean (both mean = 1.1 ng m$^{-3}$) (Gonzalez-Gaya et al., 2016). The levels of $\sum_{18}$ PAH $C_A$ were measured from North Pacific towards the Arctic Ocean with the range from 0.0002 to 0.36 ng m$^{-3}$, with the highest concentration found in the coastal areas in East Asia (Ma et al., 2013). These concentrations were significantly lower than the averages levels found in our study. Besides, Ma et al. (2013) observed the relatively higher $\sum_{18}$ PAH $C_A$ in the most northern latitudes of the Arctic Ocean, which is associated with back trajectories of air masses from Sothern Asia. The higher levels of $C_A$ in our study could be attributed to the costal line close to larger burning taiga forest and more industrial sources in the boreal regions of Russian continent. Similar to the pattern for heavy metals mentioned above, high levels of these chemicals may have been derived from atmospheric transport from the industrial areas of the Russian continent. Because different sampling methods, different measured total PAH species and not all reports separated gas and particles concentrations, it is quite difficult to compare PAH levels in the aerosols.".

As for black carbon concentration in aerosol, we did not establish the filter-based techniques or direct techniques to measure them in our study. From other studies (**Figures**), we could see that Concentrations are low in the Arctic compared with lower latitudes and come mostly from outside the area and there are lots of black carbon emission which can be transported to the Arctic regions. We think the connections between black carbon and PAHs contents can be further studied.

[Figure]

Sources: MACEB project (www.maceb.fi), IIASA-GAINS model.

[Figure]

Source: UNEP/WMO 2011. United Nations Environment Programme , World Meteorological Organization (WMO), Integrated Assessment of Black Carbon and Tropospheric Ozone. 2011. http://2011_integrated-assessment-SUMMARY_UNEP-WMO.pdf

[Figure]

Source: Koch, D et al. Corrigendum to "Evaluation of black carbon estimations in global

aerosol models" published in Atmos. Chem. Phys., 9, 9001-9026, 2009, Atmos. Chem. Phys., 10, 79–81, https://doi.org/10.5194/acp-10-79-2010, 2010.

[Figure]

Source: AMAP, 2011. The Impact of Black Carbon on Arctic Climate (2011). By: P.K. Quinn, A. Stohl, A. Arneth,T. Berntsen, J. F. Burkhart, J. Christensen, M. Flanner, K. Kupiainen, H. Lihavainen, M. Shepherd, V. Shevchenko,H. Skov, and V. Vestreng. https://www.amap.no/documents/download/977/inline

15. Generally, compare the pahs and metal concentrations with other reports for the Arctic, even if these are for the north atlantic and north pacific.
**Response**: we will add the comparisons for both metals and PAHs with other oceans.

16. Line 391. Here or in the methods, comment the range of estimated dry deposition velocities.
**Response**: dry deposition velocities ($v_D$) is very crucial factor to calculate dry deposition as shown in equation. We will add the information to describe and compare the estimated dry deposition velocities in our study.

$F_{DD}=864v_DC_A$

17. Line 395. Huge values, If possible compare with other measures. Report as well the concentrations in snow.
**Response**: we will add the content to compare with other measurements.

18. Line 398. Comment the range of diffusive fluxes, and show them in a figure, maybe in the supplementary material.
**Response**: we will add a graph and a discussion with specific diffusive fluxes of $F_{AW}$ in the current range.

19. Line 399 and figure 6b. This is not clear to me. For which pahs there is a net volatilization and for which there is a net deposition.

**Response**: we are sorry for this confusion. Actually, it is not volatilization according to the calculated $F_{AW}$, while we did not make the direction downwards. We will revise this graph.

20. Line 413. Rewrite. As I understand them, these fluxes for the the basins studied, but not all the Arctic, which should be clarified. Provide the surface for each basin in methods.
**Response**: these fluxes were based on each basin of the Russian Arctic Oceans. We will rewrite this part for better clarification for the differences between each basin and the whole Arctic oceans. The surface for each basin was based on the sampling sites during the cruise and we will add this information to the methods.

21. Review the spelling of pah's names in figure 4 and 5. The size of the legend (scale) in the figures should be bigger.
**Response**: we will remake Figure 4 and 5 for lager scale of the legend.

22. Review the use of English in the manuscript.
**Response**: after the revision of our manuscript, we will ask a native speaker to check the language thoroughly for English language in our manuscript.

23. Table S4. Why dibenzothiphene and anthracene appear twice in this table, while phenanthrene is not there? Review all the tables and data set. In addition, the mean total concentrations seem to not correspond to the distribution seen in Figure 5.
**Response**: thank you very much for your careful checking. Indeed, we have found that PAHs did not match the concentration data. We have checked our original dataset and have revised all tables. We also checked whether the data used in Figures are correct as shown in the tables.

24. Figure S5. The three legends appear as Cg… correct.
**Response**: thank you for this careful check! we will correct this graph.

25. Improve the quality of the figures in the supplementary material. Generally, the presentation aspects of this work need to be reviewed.
**Response**: we will make the figures in supplementary material easier to watch with better resolution. Thank you again for these helps. We hope you would continue helping us to revise our manuscript.

---

## Author Comment (AC3) · 24 Sep 2019

Response: Thank you for your interest in our manuscript. We will add these references recommended by you for comprehensive discussion. Thank you for pointing the problems in Figure 6, we will revise the direction of bar in Figure 6.

---

## Author Comment (AC4) · 25 Sep 2019

Dear authors, your paper about air-ocean exchange of PAHs and heavy metals is interesting. I believe the data can be fill the gap of net input/output of these substances originating from both anthropogenic sources and natural sources. The clarification and structure of the paper is clear and generally good. I have some questions concerning to this paper.

Response: thank you for reading our manuscript. We have answered your questions item by item as follows:

1. How far the samples were collected between two sample sites?

Response: because the sampling for air and water (two portions, the same amount of water was directly extracted by SPE) were carried out simultaneously, we counted the time to next destination which was determined previously according to the potential costal emission sources from the continent. According to geographic coordinate system, approximately 108 km for each sampling site on average.

2. How authors choose these sites?

Response: we separated the Russian Arctic Ocean by four seas which represent the close distance to anthropogenic activities and more natural Arctic systems.

3. Authors mentioned the atmospheric mercury depletion events (AMDEs) of Hg in the paper, so were there any sites collected during the sunrise in spring season?

Response: due to this special accumulation of Hg deposition during spring season, we have prevented sampling during the sunrise.

4. Do authors try to avoid this time?

Response: yes, we avoid this time for preventing abnormity data.

5. Were Dissolve Organic Carbon (DOC) and the concentration of surface chlorophyll same sample as the water?

Response: yes, we measured them from the same amount of water once we sampled during expedition.

6. How authors can make sure the corresponding the water sample to air samples?

Response: because the huge time gap and distance between two samples' site, we think the relatively error for this larger site is not very high.

7. Authors calculated the atmospheric oxidation of PAHs initiated by reaction with the OH radical in gas phase?

Response: yes, because previous studies have shown that in the gas phase naphthalene reacts with OH radicals and $NO_3$ radicals by initial addition of these radicals to the aromatic rings. he gas-phase OH radical and $NO_3$ radical reactions have been shown to proceed by initial addition to the aromatic rings to form hydroxycyclohexadienyl- and nitrooxycyclohexadienyl-type radicals, which can back-decompose to the reactants or react with $NO_2$ or $O_2$ to yield products. I listed some relevant references as shown below:

Jin Shi, Wenlong Bi, Shenmin Li, Wenbo Dong, and Jianmin Chen . Reaction Mechanism of 4-Chlorobiphenyl and the NO3 Radical: An Experimental and Theoretical Study. The Journal of Physical Chemistry A 2017, 121 (18), 3461-3468. DOI: 10.1021/acs.jpca.6b08626.

Matthieu Riva, Robert M. Healy, Pierre-Marie Flaud, Emilie Perraudin, John C. Wenger, and Eric Villenave . Gas- and Particle-Phase Products from the Chlorine-Initiated Oxidation of Polycyclic Aromatic Hydrocarbons. The Journal of Physical Chemistry A 2015, 119 (45), 11170-11181. DOI: 10.1021/acs.jpca.5b04610.

Hyun Ji (Julie) Lee, Paige Kuuipo Aiona, Alexander Laskin, Julia Laskin, and Sergey A. Nizkorodov . Effect of Solar Radiation on the Optical Properties and Molecular Composition of Laboratory Proxies of Atmospheric Brown Carbon. Environmental Science & Technology 2014, 48 (17), 10217-10226. DOI: 10.1021/es502515r.

Abolfazl Shiroudi, Michael S. Deleuze, and Sébastien Canneaux . Theoretical Study of the Oxidation Mechanisms of Naphthalene Initiated by Hydroxyl Radicals: The OH-Addition Pathway. The Journal of Physical Chemistry A 2014, 118 (26), 4593-4610. DOI: 10.1021/jp411327e

8. Is it possible to be also degradable in the particle phase?

**Response**: it is also possible. However, we took account into only the degradation of gas phase PAHs, ignoring the potential degradation of aerosol-bound PAHs due to these sources of uncertainty, the figures given here for degradative fluxes have an error of factor of one to three depending on the individual PAH. We do not provide the atmospheric degradation fluxes since major uncertainties in their $k_{OH}$ values.

---

## Author Response (AR1)

**From Anonymous Referee #1**

The MS titled "Atmosphere–ocean exchange of heavy metals and polycyclic aromatic hydrocarbons in the Russian Arctic Ocean" written by Ji et al. researched the results of a Russian Arctic assessment of the occurrences and atmosphere–ocean fluxes of 35 polycyclic aromatic hydrocarbons (PAHs) and 9 metals (Pb, Cd, Cu, Zn, Fe, Mn, Ni, and Hg). The topic of this study is quite interesting due to the reflection of anthropogenic influences in the normal biochemical cycles as the balance of geochemical substances between ocean and gas. Authors pointed out the net input of Hg and 35 PAHs into ocean, filling the data gaps in this field. The deposition including dry and wet deposition in the Arctic Ocean so far appeared very sporadic data without a continuous sampling sites and period. The topic of this MS is within scope of ACP, the language usage and structure of article is generally good and well-written. However, there are still some minor problems and some questions remained. I would like to support this MS to be published unless my questions and problems are well addressed or answered. There are specific questions I found in this MS.
**Response**: thank you very much for your valuable and helpful comments and suggestions on our manuscript.

1. L43-44, it's quite not logic here, authors should introduce the transported pollutants from low attitude to the polar region as the Arctic rather than the present statements about reducing global emissions. Additionally, global emissions of the atmospheric pollutants were carried by monsoon to the high latitude. What is the contribution of it to the Arctic area? Are there any reports concerning the separate contribution of airborne pollutants to land and ocean?
**Response**: initially, we tried to show Arctic air pollution includes harmful trace gases (e.g. tropospheric ozone) and particles (e.g. black carbon, sulphate) and toxic substances (e.g. polycyclic aromatic hydrocarbons) that can be transported to the Arctic from emission sources located far outside the region, or emitted within the Arctic from activities including shipping, power production, and other industrial activities **(Arnold et al. 2015)**. We realized it's not appropriate to put global emissions here with a specific contribution to the Arctic. We will put more references about Arctic air pollutants' transport for better illustration here.

Arnold, S.R., Law, K.S., Brock, C.A., Thomas, J.L., Starkweather, S.M., Salzen, K. von ., Stohl, A., Sharma, S., Lund, M.T., Flanner, M.G., Petäjä, T., Tanimoto, H., Gamble, J., Dibb, J.E., Melamed, M., Johnson, N., Fidel, M., Tynkkynen, V.-P., Baklanov, A., Eckhardt, S., Monks, S.A., Browse, J. and Bozem, H., 2016. Arctic air pollution: Challenges and opportunities for the next decade. Elem Sci Anth, 4, p.000104. DOI: http://doi.org/10.12952/journal.elementa.000104

2. L46, the reason mercury is key problematic pollutant should be briefly mentioned here. And citation is needed here.
**Response**: we have revised this sentence as "mercury is a key problematic pollutant in the Arctic because mercury is a neurotoxic pollutant seriously influencing northern latitudes through human exposure originated from eating seafood and marine mammals as traditional diets by hunting and fishing (Stow et al., 2015)" in **line 46-48, Page 3**.

L47, what kind of sources are referring here?
**Response**: the sources could be as the resulting sea ice loss may increase accessibility of the Arctic, leading to increases in air pollutant emissions within the Arctic from activities such as oil and gas extraction or shipping. It is thought that Northern Hemisphere mid-latitude emissions (from Europe, Asia, and North America) are currently the main source of air pollutants in the Arctic (**Stohl, 2006; Sharma et al., 2013**), including also toxic contaminants with important atmospheric pathways (e.g. mercury (Hg), certain persistent organic pollutants (POPs). However, sources of air pollution from within the Arctic or nearby sub-Arctic (defined here as 'local') are already important in some regions **(Stohl et al. 2013)**, and these and other sources may grow rapidly in the future **(Corbett et al., 2010; Peters et al., 2011)**. We will revise this part of information.

Shindell D, Faluvegi G. 2009. Climate response to regional radiative forcing during the twentieth century. Nat Geosci 4: 294–300. doi: 10.1038/ngeo473.

Stohl A. 2006. Characteristics of atmospheric transport into the Arctic troposphere. J Geophys Res 111: D11306. doi: 10.1029/2005JD006888.

Sharma S , Ishizawa M , Chan D , Lavoue D , Andrews E , et al. 2013. 16-Year simulation of Arctic black carbon: Transport, source contribution, and sensitivity analysis on deposition. J Geophys Res 118: D017774. doi: 10.1029/2012JD017774.

Stohl A , Klimont Z , Eckhardt S , Kupiainen K , Shevchenko VP , et al. 2013. Black carbon in the Arctic: The underestimated role of gas flaring and residential combustion emissions. Atmos Chem Phys 13: 8833–8855. doi: 10.5194/acp-13-8833-2013.

Peters G , Nilssen T , Lindholt L , Eide M , Glomsrød S , et al. 2011. Future emissions from shipping and petroleum activities in the Arctic. Atmos Chem Phys 11: 5305–5320. doi: 10.5194/acp-11-5305-2011.

Corbett JJ , Lack DA , Winebrake JJ , Harder S , Silberman JA , et al. 2010. Arctic shipping emissions inventories and future scenarios. Atmos Chem Phys 10: 9689–9704. doi: 10.5194/acp-10-9689-2010.

3. L49, "the melting of contaminated ice" in the ocean or also in the terrestrial land?
**Response**: In here, we meant both melting ice in ocean as well as the melting snow in the terrestrial soils since fluxes from thermokarst rivers would bring the pollutants into the ocean as another source.

4. L68, what is relevant connection between benthic input of metals and air-seawater exchange input?
**Response**: It has been reported that a large fraction of the organic matter that forms in surface waters in the shelf areas of the Chukchi Sea sinks to the sea floor, which fuels productive

benthic communities and causes high rates of sedimentary denitrification **(Chang and Devol, 2009; Brownetal., 2015)**. The Pacific origin water from the Bering Strait is already deplete dinitrate relative to phosphate, and $NO_3^-$ is further depleted relative to $PO_4^{3-}$ in the Chukchi Sea via the effect of sedimentary denitrification **(Yamamoto-Kawaietal.,2006)**. A unique feature of the upper surface water in the western Arctic Ocean is the dominance of a strong, cold halocline that separates the Pacific-origin surface waters from the underlying Atlantic-origin waters **(Aagaardetal.,1981)**. And metals in deep ocean showed the similar pattern of nutrients (**Brownetal., 2015)**.

Chang, B.X., Devol,A.H., 2009. Seasonal and spatial patterns of sedimentary denitrification rates in the Chukchi sea. Deep Sea Res. II56(17), 1339–1350.

Yamamoto-Kawai, M., Carmack, E., McLaughlin, F., 2006. Nitrogen balance and Arctic through flow. Nature 443,43.

Aagaard, K., Coachman,L.K.,Carmack,E., 1981.On the halocline of the Arctic Ocean. Deep Sea Res. 28A(6), 529–545.

Brown,Z.W., Casciotti,K.L., Pickart,R.S., Swift,J.H., Arrigo,K.R., 2015.Aspects of the marine nitrogen cycle of the Chukchi Sea shelf and Canada Basin. Deep Sea Res. II 118,73–87.

5. L88-89, the sentence authors make here tried to indicate the inorganic salt ions may increase during the summer melting season, and thence the organic compounds' solubility would change? Now the meaning is not clear. Authors need to rephase the sentences to clarify.
**Response**: we will make this sentence clear. We intended to state that inorganic salts are appreciably soluble in organic media, principally of the oxygenated forms (alcohols, ketones, ethers, esters), when they are capable of forming neutral molecules in solution. This property is largely confined to the transition elements. Those of the first long series, and in the trans-radium region, accomplish this without forming undebatably covalent compounds; other subgroup elements which are solvent-soluble show a stronger tendency toward covalent bond formation, and may show solubility in chlorinated hydrocarbons and benzenoid solvents in which the first group mentioned are not usually soluble. The salts of the first three groups of the periodic table generally are not organic-soluble, because their co-ordinative power is relatively low, and their solid lattice energies are high.

6. L91, need citation for "the Arctic Ocean is considered as a sink that receives global airborne pollutants".
**Response**: we have added a reference for the source of this sentence in **line 87, Page 4**.

7. L203, how surface chlorophyll concentrations were calculated or measured?
**Response**: Spectrophotometry method was used to measure chlorophyll concentrations. It involves the collection of a fairly large water sample, filtration of the sample to concentrate the chlorophyll-containing organisms, mechanical rupturing of the collected cells, and

extraction of the chlorophyll from the disrupted cells into the organic solvent acetone. The extract is then analyzed by either a spectrophotometric method (absorbance or fluorescence), using the known optical properties of chlorophyll, or by HPLC. This general method, detailed in Section 10200 H. of Standard Methods, has been shown to be accurate in multiple tests and applications and is the procedure generally accepted for reporting in scientific literature.

8. L218, why H' can be corrected by the salinity? In Fick's law, I am not aware of such connection. H' values are usually considered by the temperature changes.
**Response**: usually temperature an salinity can be both used to correct Harry laws. However, the salinity of seawater has been indirectly determined by means of electrical conductivity. Since the absolute conductivity cannot be measured as accurately as required for precise salinity measurements **(Seitz et al., 2010)**, the conductivity has been measured relative to that of standard seawater; the conversion to salinity is carried out by means of the (relative) conductivity–salinity relation PSS-78 **(JPOTS, 1981a, b)**. In practice, this is achieved by calibrating salinometers and conductivity–temperature–depth devices using standard seawater, which is diluted to obtain the conductivity of the potassium chloride standard **(Culkin, 1986; Bacon et al., 2007)** used as a conductivity reference. An unconditional prerequisite for the comparability of salinity measurements over long periods is, therefore, that the salt proportions in standard seawater are stable. Unfortunately, this cannot be guaranteed, as standard seawater is of natural origin.

Seitz, S., Spitzer, P., and Brown, R. J. C.: CCQM-P111 study on traceable determination of practical salinity and mass fraction of major seawater components, Accredit. Qual. Assur., 15, 9–17, https://doi.org/10.1007/s00769-009-0578-8, 2010.

Joint Panel on Oceanographic Tables and Standards (JPOTS): Tenth report of the Joint Panel on Oceanographic Tables and Standards – The Practical Salinity Scale 1978 and The International Equation of State of Seawater 1980, Unesco technical papers in marine science, 36, 13–17, UNESCO, Paris, France, available at: http://unesdoc.unesco.org/images/0004/000461/046148eb.pdf , 1981a.

Joint Panel on Oceanographic Tables and Standards (JPOTS): Background papers and supporting data on the Practical Salinity Scale 1978, Unesco technical papers in marine science, 37, UNESCO, Paris, France, available at: http://unesdoc.unesco.org/images/0004/000479/047932eb.pdf , 1981b.

Bacon, S., Culkin, F., Higgs, N., and Ridout, P.: IAPSO standard seawater: definition of the uncertainty in the calibration procedure and stability of recent batches, J. Atmos. Ocean. Tech., 24, 1785–1799, https://doi.org/10.1175/JTECH2081.1, 2007.

Culkin, F.: Calibration of standard seawater in electrical conductivity, Sci. Total Environ., 49, 1–7, https://doi.org/10.1016/0048-9697(86)90230-5, 1986.

9. L246, how the uncertainty of air-water exchange net direction was conducted?

**Response**: we will add this information to the supplementary information. We take the method from Liu et al. (2016).

Liu, Y., Wang, S., McDonough, C. A., Khairy, M., Muir, D., and Lohmann, R.: Estimation of Uncertainty in Air–Water Exchange Flux and Gross Volatilization Loss of PCBs: A Case Study Based on Passive Sampling in the Lower Great Lakes, Environmental Science & Technology, 50, 10894-10902, 10.1021/acs.est.6b02891, 2016.

To evaluate the uncertainty in air-water fugacity ratio and the statistically calculated diffusive flux, measured uncertainties of water and air analysis, Henry's law constant, temperature and overall velocity of mass transfer were taken into account. Four variables with random uncertainty of the fugacity ratio was based on Eq. (11) and Eq. (13), of which the uncertainty is shown in Eq. (S1).

$$\delta \ln\left(\frac{f_g}{f_w}\right) = \sqrt{\left(\frac{\delta C_g}{C_g}\right)^2 + \left(\frac{\delta C_w}{C_w}\right)^2 + \left(\frac{\delta H'}{H'}\right)^2 + \left(\frac{\delta T}{T}\right)^2} \quad \text{(S1)}$$

The relative standard deviation (RSD) of aqueous and water concentrations ($\left(\frac{\delta C_g}{C_g}\right)$ and $\left(\frac{\delta C_w}{C_w}\right)$) are relevant to the analysis. The RSDs of H' was taken value as 50%.

10. L263, why only OH radicals were considered to the degradation of PAHs in the atmosphere?

**Response**: Dry deposition is more effective than wet deposition as a removal process from the atmosphere. Chemical reactions provide the other main sink for atmospheric PAHs. The gas phase reactions of PAHs with the OH radical, the $NO_3$ radical and ozone have been widely investigated. Available rate coefficient data are most abundant in the case of the OH radical. The established mechanism of PAH reactions with the OH radical involves the formation of a PAH–OH adduct followed by further reaction with $NO_2$ or $O_3$. The observed reaction products include both ring-retaining nitro-PAHs and quinones, as well as ring-opened products such as phthalic acid, phthalaldehyde and phthalic anhydride. The presence of methyl groups in methyl naphthalenes and methyl phenanthrenes in most cases leads to a modest increase in reactivity relative to the parent PAH. For NO3 reactions, the predominant reaction pathway involves NO3 addition followed by reaction with $NO_2$ leading to nitro-PAH formation. The observed rate coefficients are proportional to the nitrogen dioxide concentration. There have been far fewer studies of the gas phase reactions of PAH with ozone.

11. L279-280, Do author have any proofs to support the reason with air trajectories of Russian Arctic?

**Response**: this reason with air trajectories of Russian Arctic was observed by the previous study **(Shevchenko et al. 2003)**.

V. Shevchenko, A. Lisitzin, A. Vinogradova, R. Stein. Heavy metals in aerosols over the seas of the Russian Arctic. The Science of the Total Environment 306 (2003) 11–25.

12. L294, which sea Taymyr Peninsula is closer to? Clarify here. L295, I think Shevchenko et al. 2013 should be moved to previous sentence.
**Response**: this phase has been revised as "On the Taymyr Peninsula (adjacent to Kara Sea and Leptev Sea)" in **line 301 Page 11**. The sentence of Shevchenko et al. 2013 has been moved to previous sentence.

L419, Can authors explain what molecular weight PAHs were easier to de degraded in the air because it's important for biocycles to consume these carbon sources by organisms. If most of heavy molecular PAHs enter ocean ecosystem, which may not be consumed by biota.
**Response**: theoretically, lighter PAHs can be easier degraded. Photo-induced toxicity of PAHs can be driven from formation of intracellular singlet oxygen and other reactive oxygen species (ROS) that cause oxidative damage in biological systems **(El-Alawi et al., 2002)**, or formation of photo-products, which exert different, often stronger, bioactivity than the parent compound **(Grote et al., 2005)**.

Measurement of short- and long-term toxicity of polycyclic aromatic hydrocarbons using luminescent bacteria. El-Alawi YS, McConkey BJ, George Dixon D, Greenberg BM Ecotoxicol Environ Saf. 2002 Jan; 51(1):12-21.

Modeling photoinduced algal toxicity of polycyclic aromatic hydrocarbons. Grote M, Schüürmann G, Altenburger R Environ Sci Technol. 2005 Jun 1; 39(11):4141-9.

L440, I suggest authors to put some information of future protective measures for this region.
**Response**: according to your suggestion, we have added the future protective measures for this region in the end of conclusion part.

**From Anonymous Referee #2**

Review of "Atmosphere-ocean Exchange of heavy metals and polycyclic aromatic hydrocarbon in the Russian Arctic Ocean".

This work reports the concentrations of PAHs and metals in air, water and snow in the Russian sector of the Arctic. There is no previous data for this sector, so this contribution is very important. The work is of mix quality, with sections that are generally well done, but other material that is erroneous or needs further work. The manuscript needs some work in order to present the data appropriately, improve the comparison with other studies for polar regions, and give some extra depth to the discussion. This revision is mainly for the PAHs part. I suggest moderate to major modifications before it can be accepted.
**Response**: we are very grateful that you spent your valuable time reviewing our manuscript and giving us good comments and suggestions to help us revise this manuscript!

1. Line 40. Comment and cite a work on long-range atmospheric transport of PAHs (or pops) to the arctic.
**Response**: according to the reviewer's suggestion, a citation about 20 years monitoring POPs in the Arctic (**Hung et al. 2016**), has been added in **Line 39 Page 3**.

Hung, H., Katsoyiannis, A. A., Brorström-Lundén, E., Olafsdottir, K., Aas, W., Breivik, K., Bohlin-Nizzetto, P., Sigurdsson, A., Hakola, H., Bossi, R., Skov, H., Sverko, E., Barresi, E., Fellin, P., and Wilson, S.: Temporal trends of Persistent Organic Pollutants (POPs) in arctic air: 20 years of monitoring under the Arctic Monitoring and Assessment Programme (AMAP), Environ. Pollut., 217, 52-61, https://doi.org/10.1016/j.envpol.2016.01.079, 2016.

2. Line 49-50. Provide examples, there are some published for Antarctica and the Arctic for both metals and pahs.
**Response**: we have carefully searched Web of Science. Some relevant references concerning PAHs and heavy metals in air deposition and sea in the Arctic that we have added as examples in **Line 49-57 Page 3**.

3. Note that not all PAHs are persistent, it also depends if they are found in the gas or dissolved phase (less persistent) or associated to aerosols and particulate matter.
**Response**: thank you for pointing out this important note. In the examples we added, we have clarified that aerosols contained PAHs deposited into sea ice and soils (or snow) which is an important source for the contribution of persistent PAHs. Indeed, according to the previous study (**Fernandes and Sicre, 1999**), the major PAHs either from adjacent areas or low latitudes are in the form of aerosols and binding with black carbon (particulate matter).

Fernandes, M. B., and Sicre, M. A.: Polycyclic Aromatic Hydrocarbons in the Arctic: Ob and Yenisei Estuaries and Kara Sea Shelf, Estuarine, Coastal and Shelf Science, 48, 725-737, https://doi.org/10.1006/ecss.1999.0472, 1999.

4. Line 73. Biogeochemical instead of biochemical
**Response**: “biochemical” has been changed to “biogeochemical” in **Line 80 Page 4**.

5. Line 77-79. Specify if you are referring to rain or snow, or both. I guess snow.
**Response**: we have specified the wet deposition as snow deposition in **Line 87 Page 4**.

6. Line 110. This citation is not adequate here.
**Response**: we realized the methods here concerning simultaneous sampling both for gas and aerosol phase PAHs. The citations of details about these two methods have added in Line

7. Line 116. I guess that first in aluminum foil and after in polyethylene bags, which must be air-tight.
**Response**: after air sampling in the ship, we did kept the samples in folded aluminium foil firstly. This sentence has been revised as “the filters and PUFs were firstly covered with aluminum foil tightly for air-tightness, then immediately placed in polyethylene bags and zip the bags, and frozen at -20 °C prior to chemical analyses.” in **Line 122-123 Page 6**.

8. Line 119. Was snow melted immediately? How was snow melted?
**Response**: no, snowfall samples were not melted immediately. After sampling, all samples were protected from light and stored at 4 °C in borosilicate glass bottles prior to the analysis. Snowfall samples were melted thoroughly at room temperature. This information has been added in **Line 130 Page 6**.

9. Line 123. M3 or L, these are huge volumes. I have never seen such large volumes for water using a XAD. Unless there is a typo mistake, such volumes need a justification and discussion.
**Response**: Thank you for pointing out this mistake. The unit should be “mL”. This mistake has been corrected in **Line 130 Page 6**. We only used XAD-2 resin rather than XAD-4 resin for high volume of water. Water was filtered at a flow rate averaging 4 L/min with a pneumatic pump (Flojet) through a borosilicate microfiber glass filters (1 mm nominal pore size), housed in an aluminum filter support.

10. Line 265. It cannot be equation 13.
**Response**: we are sorry for this mistake. The hydroxyl radicals concentrations [OH] in the considered mixed layer (between 1,000 and 500 hPa) was based on the zonally and monthly averaged concentrations of OH radicals from **Spivakovsky and coworkers (2000)**. Therefore, this sentence has been removed.

Spivakovsky, C. M. et al. Three-dimensional climatological distribution of tropospheric OH: update and evaluation. J. Geophys. Res. 105, 8931-8980 (2000).

11. Line 266. Is this concentration appropriate for the arctic? Discuss.
**Response**: This is a very good question to discuss! We must admit that there is no ranges or exact data of OH radical levels in troposphere of polar regions.

Firstly, the primary source of HO radical in the troposphere is the photolysis of $O_3$ to produce O ($^1D$). This reaction requires radiation of wavelength less than 315nm, otherwise ($O^3P$) is produced. The latter species reacts with molecular oxygen in the presence of $N_2$ or $O_2$ to produce $O_3$. The electronically excited O($^1D$) usually relaxes to produce ($O^3P$) but also undergoes reaction with water to produce HO. While minor sources of HO are available through reaction of O($^1D$) with $CH_4$ and $N_2$. Therefore, the methods to measure OH concentrations were usually based on observed distributions of $O_3$, $H_2O$, $NO_t$ ($NO_2$+NO+$2N_2O_5$+$NO_3$ +$HNO_2$+$HNO_4$), CO, hydrocarbons, temperature, and cloud optical depth.

Thus, it is difficult to compare the OH concentrations by different methods and altitudes as well as latitudes. **Hewitt and Harrison (1985)** summarized the OH concentrations in different altitude as well as the global mean, which showed the range of $0.5\text{-}5 \times 10^6$ mol $cm^{-3}$ for daytime OH radical. **Li et al. (2008)** used an empirical method is presented to determine effective OH concentrations in the troposphere and lower stratosphere, based on $CH_4$, $CH_3Cl$, and $SF_6$ data from aircraft measurements (IAGOS-CARIBIC) and a ground-based station (NOAA). The results showed tropospheric OH average values of $10.9 \times 10^5$ ($\sigma = 9.6 \times 105$) mol $cm^{-3}$ in a global level. The reason we used data from **Spivakovsky and coworkers (2000)** is that the estimation of OH concentrations considered by latitudes ($\pm 32^o$) in Northern Hemisphere.

It should be noted that precipitation amounts in the Arctic are very low (annual precipitation on Svalbard is 150-300 mm). This means that wet scavenging is not efficient and the lifetime for soluble species like aerosols is longer than on the continents to the south. Gas- phase chemical removal of trace compounds all but stops in the Arctic atmosphere during the polar night. In the absence of sunshine, the production rate of the OH radical, which is the main gas-phase scavenger, is low. Also, in the sunlit part of the year, the chemical lifetime of trace compounds is relatively long in the polar atmosphere (except in the near-surface layer that is impacted by snowpack photochemical processes), due to the strong attenuation of short wave visible sunlight as the solar elevation is low and the low specific humidity. Therefore, the OH radicals in the Arctic troposphere should be further measured throughout the year. So far, OH radicals levels are within the global levels.

C.N. Hewitt, Roy M. Harrison, Tropospheric concentrations of the hydroxyl radical—a review,
Atmospheric Environment (1967), Volume 19, Issue 4, 1985, Pages 545-554.

Li, Mengze; Karu, Einar; Brenninkmeijer, Carl; Fischer, Horst; Lelieveld, Jos; Williams, Jonathan; 2018; npj Climate and Atmospheric Science, (1): 29.

Spivakovsky, C. M. et al. Three-dimensional climatological distribution of tropospheric OH: update and evaluation. J. Geophys. Res. 105, 8931-8980 (2000).

12. Line 269-270. I cannot understand this sentence if these estimations have just been explained.

**Response**: the uncertainty (error propagation) analysis basically used the relative standard deviation (RSD) based on measured uncertainties in air and water analysis, air−water partitioning coefficients (including Henry's law constant and temperature), and overall mass transfer velocity were considered. The equation we have added in **Text S1 Supplementary Materials**.

13. Line 370. In addition to the highest, provide the range for each basin. Generally, the values commented here do not correspond always to the values seen in Figure 5.

**Response**: the range of $\sum_{35}$ PAH concentrations in gas, aerosols and dissolve water for each basin has been added in **Line 373-397 Page 14-15**. We have checked the original data in Figure 5, some wrong concentration values have been revised in this paragraph (**Line 379-395, Page 14**).

14. Line 372. The concentrations in aerosols are extremely high! These need a comparison with other studies and a discussion. Which is the black carbon concentration in aerosol for this region?

**Response**: we are very sorry that this concentration was showing wrong in the paper. It cannot be so high. All concentrations should be divided 100. We have made corrections for graphs and texts. There are some reports concerning PAH concentrations in atmosphere in Ocean. However, not all separating gas and particle phase. Therefore, we only compared and discussed with the separated ones as "$\sum_{35}$ PAH concentrations in aerosols ($C_A$, ng $m^{-3}$) in the Barents Sea (0.25-2.95), and East Siberian Sea (0.24-3.32) with average $C_A$ values of 1.38 and 2.07 ng $m^{-3}$ respectively were apparently higher than those in the Leptev Sea (0.23-0.89) and Kara Sea (0.23-0.27) with average CA values of 0.30 and 0.25 ng $m^{-3}$, respectively (Fig. 5b). The average CA of $\sum_{35}$ PAH in present study is higher than those of $\sum_{64}$ PAH measured in South Atlantic Ocean (mean = 0.93 ng $m^{-3}$) and North Pacific Ocean (mean = 0.56 ng m-3) while much lower than those of $\sum_{64}$ PAH in Indian Ocean (mean = 10 ng m-3) (Gonzalez-Gaya et al., 2016). Considering average $\sum_{35}$ PAH $C_A$ (1.02 ng $m^{-3}$) in the Russian Arctic Ocean, the value is comparable to those of $\sum_{64}$ PAH observed in South Atlantic Ocean and South Pacific Ocean (both mean = 1.1 ng $m^{-3}$) (Gonzalez-Gaya et al., 2016). The levels of $\sum_{18}$ PAH $C_A$ were measured from North Pacific towards the Arctic Ocean with the range from 0.0002 to 0.36 ng $m^{-3}$, with the highest concentration found in the coastal areas in East Asia (Ma et al., 2013). These concentrations were significantly lower than the averages levels found in our study. Besides, Ma et al. (2013) observed the relatively higher $\sum_{18}$ PAH $C_A$ in the most northern latitudes of the Arctic Ocean, which is associated with back trajectories of air masses from Sothern Asia. The higher levels of $C_A$ in our study could be attributed to the costal line close to larger burning taiga forest and more industrial sources in the boreal regions of Russian continent. Similar to the pattern for heavy metals mentioned above, high levels of these chemicals may have been derived from atmospheric transport from the industrial areas of the Russian continent. Because different sampling methods, different measured total PAH species and not all reports separated gas and particles concentrations, it is quite difficult to compare PAH levels in the aerosols." In **Line 376-394 Page 14**.

As for black carbon concentration in aerosol, we did not establish the filter-based techniques or direct techniques to measure them in our study. From other studies (**Figures**), we could see that Concentrations are low in the Arctic compared with lower latitudes and come mostly from outside the area and there are lots of black carbon emission which can be transported to the Arctic regions. We think the connections between black carbon and PAHs contents can be further studied.

[Figure]

Sources: MACEB project (www.maceb.fi), IIASA-GAINS model.

[Figure]

Source: UNEP/WMO 2011. United Nations Environment Programme , World Meteorological Organization (WMO), Integrated Assessment of Black Carbon and Tropospheric Ozone. 2011. http://2011_integrated-assessment-SUMMARY_UNEP-WMO.pdf

[Figure]

Source: Koch, D et al. Corrigendum to "Evaluation of black carbon estimations in global aerosol models" published in Atmos. Chem. Phys., 9, 9001-9026, 2009, Atmos. Chem. Phys., 10, 79–81, https://doi.org/10.5194/acp-10-79-2010, 2010.

[Figure]

Source: AMAP, 2011. The Impact of Black Carbon on Arctic Climate (2011). By: P.K. Quinn, A. Stohl, A. Arneth,T. Berntsen, J. F. Burkhart, J. Christensen, M. Flanner, K. Kupiainen, H. Lihavainen, M. Shepherd, V. Shevchenko,H. Skov, and V. Vestreng. https://www.amap.no/documents/download/977/inline

15. Generally, compare the pahs and metal concentrations with other reports for the Arctic, even if these are for the north atlantic and north pacific.

**Response**: we have added the comparison both for metals and PAHs in the manuscript (**Page 11, 14-15**). However, we only compared with the similar sampling methods. Some of them with other method as unit "ng/g" for air concentrations were incomparable with our study. therefore, we didn't compare these publications.

16. Line 391. Here or in the methods, comment the range of estimated dry deposition velocities.

**Response**: the range of dry deposition velocities ($V_d$) is important to calculate the dry deposition fluxes. $V_d$ for each individual PAH may vary significantly. We have added the discussion as "The increasing values of dry deposition ($V_d$) may influence $F_{DD}$ in the marine environment due to the higher hydrophobicity of organic compounds, surface microlayer with reduced surface tension, and lipid floating (del Vento and Dachs, 2007b). The apparently higher average $V_d$ was observed for 9-methlyfluorene (1.01-10.02 cm $s^{-1}$) followed by 1,7-dimethylfluorene (1.06-10.63 cm $s^{-1}$) (**Fig. S6**). In the global scale, higher $V_d$ was found for heavier PAHs such as methylchrysene (0.17-13.30 cm $s^{-1}$) and dibenzo(a,h)anthracene (0.29-1.38 cm $s^{-1}$) and other heavier PAHs (Gonzalez-Gaya et al., 2014). The $V_d$ values reported previously ranged from 0.08 to 0.3 cm $s^{-1}$ in the Atlantic Ocean (Del Vento and Dachs, 2007a), from 0.01 to 0.8 cm $s^{-1}$ coastal areas (Holsen and Noll, 1992;Bozlaker et al., 2008;Esen et al., 2008;Eng et al., 2014). The previous reports showed a higher $V_d$ values in concentrated industrial and urban areas (Bozlaker et al., 2008). In our study, the highest $V_d$ values were observed in Barents Sea and the other three seas did not show the statistical differences ($p > 0.05$) except for 9-methlyfluorene and 1,7-dimethylfluorene. East Siberian Sea showed the lowest value of $V_d$ while the relatively higher Vd values were found for heavier PAHs (dibenzo(a,h)anthracene, indeno(1,2,3-cd)pyrene, dibenzo(a,h)anthracene and benzo(g,h,i)perylene) in all the seas (**Fig. S7**). This may be explained by heavier PAHs are principally deposited via heavier aerosols with a quicker $V_d$ because of bound with hydrophobic aerosols or gravity such as soot carbon having a faster $V_d$ (Gonzalez-Gaya et al., 2014)." in **Line 309-427 Page 15**.

17. Line 395. Huge values, If possible compare with other measures. Report as well the concentrations in snow.

**Response**: we have recalculated our data and revised the values and unit. In the most references, the measurement for wet depositions was similar with use of high-volume air sampler for snow and rain, and the sample media consisted of a glass fiber filter (GFF) to trap airborne particles, followed by a self-packed PUF/XAD-2 glass column. As for papers focusing on new methods of measuring wet deposition we did not consider to compare in our paper. We have added some comparisons with other regions, however, some reference with too low wet deposition such as snowpack from lakes of Western U.S. National Parks (0.005 μg $m^{-2}$ $y^{-1}$ to 0.1μg $m^{-2}$ $y^{-1}$) (**Usenko et al. 2010**) were not considered due to too limited sources compared to our study.

Usenko, S.; Simonich, S. L. M.; Hageman, K. J.; Schrlau, J. E.; Geiser, L.; Campbell, D. H.; Appleby, P. G.; Landers, D. H. Sources and Deposition of Polycyclic Aromatic Hydrocarbons to Western U.S. National Parks. Environ. Sci. Technol. 2010, 44 (12), 4512– 4518, DOI: 10.1021/es903844n

We have added the content as "The wet deposition flux of the ∑35 PAHs ($F_{WD}$, μg m-2 d-1) ranged from 14 to 19. Gonzalez-Gaya et al. (2014) found the highest $F_{WD}$ of ∑64 PAHs in North Atlantic Ocean (24 μg $m^{-2}$ $d^{-1}$) with an average $F_{WD}$ value about 8 μg $m^{-2}$ $d^{-1}$ in the

global scale based on the rain samples. The apparent higher $F_{WD}$ values of PAHs were found in urban areas of China (62.6 μg $m^{-2}$ $d^{-1}$) (Wang et al., 2016) from rain samples and much lower $F_{WD}$ values of PAHs (0.02-0.28 μg $m^{-2}$ $d^{-1}$) from both rain and snow were observed in high mountain European areas (Arellano et al., 2018). Our $F_{WD}$ values were within the range of global scale and the difference of wet deposition was mainly depended on the source distance and precipitation intensity." in **Line 427-435 Page 15-16**.

18. Line 398. Comment the range of diffusive fluxes, and show them in a figure, maybe in the supplementary material.

**Response**: we have added the description and graph of $F_{AW}$ fluxes as "The estimated net diffusion of air–water exchange ($F_{AW}$, ng $m^{-2}$ $d^{-1}$) revealed that most PAHs had net inputs from the atmosphere to ocean except for the more volatile PAHs such as 2–3 ring PAHs (Fig. 6b). The lighter PAHs (2-3 rings) appeared more volatilization trend (978-4892 ng $m^{-2}$ $d^{-1}$) while heavier PAHs (4-6 rings) showed net deposition (1561-7808 ng $m^{-2}$ $d^{-1}$) except for dibenzo(a,h)anthracene (1322 ng m-2 d-1), indeno(1,2,3-cd)pyrene (1238 ng $m^{-2}$ $d^{-1}$), trimethylphenanthrene (1901 ng $m^{-2}$ $d^{-1}$) and benzo(g,h,i)perylene (2708 ng $m^{-2}$ $d^{-1}$), and three orders of magnitudes higher net deposition were observed for methylphenanthrene, dimethylphenanthrene, trimethylphenanthrene and tetramethylphenanthrene (Fig. S9). Our results were similar to the volatilization of costal water in other PAH-affected areas such as southeast Mediterranean (Castro-Jimenez et al., 2012), Narragansett Bay (Lohmann et al., 2011) and North Atlantic Ocean (Lohmann et al., 2009). Ma et al. (2013) suggested that slight volatilization of lighter PAHs may exist additional sources as ship ballast and riverine runoff, which was consistent with our study that the higher volatilization was found in East Siberian Sea and Barents Sea where more industrial factories and urban areas are situated. Our study is also consistent with previous reports in which the results showed that diffusion during air–water exchange is the main process for transfers of relatively lighter volatile organic compounds in the marine environment (Castro-Jimenez et al., 2012;Jurado et al., 2005)." in **Line 436-449 Page 16**.

[Figure]

**Figure S9.** Estimated average net diffusive air-water exchange (FAW) of PAHs in the Russian Arctic Ocean. Bars represent standard deviation.

19. Line 399 and figure 6b. This is not clear to me. For which pahs there is a net volatilization and for which there is a net deposition.

**Response**: we have remade these figures as Figure 6-7.

20. Line 413. Rewrite. As I understand them, these fluxes for the the basins studied, but not all the Arctic, which should be clarified. Provide the surface for each basin in methods.

**Response**: we have rewritten this sentence as "This indicates that atmospheric transport of PAHs derived from anthropogenic activities is for all sectors of the Russian Arctic Ocean while only East Siberian Sea and Leptev Sea have more anthropogenic PAHs in water phase." **in line 476-478, Page 17**. We did not divide the four seas very specifically since we sampled based on ship route and we have added the island marks to divided in surface for each sea basin in the methods **in Line 113-114, Page 5**.

21. Review the spelling of pah's names in figure 4 and 5. The size of the legend (scale) in the figures should be bigger.

**Response**: we have checked the spelling and remade the Figures to make the legend larger.

22. Review the use of English in the manuscript.

**Response**: we have asked a native speaker to check the language throughout the manuscript.

23. Table S4. Why dibenzothiphene and anthracene appear twice in this table, while phenanthrene is not there? Review all the tables and data set. In addition, the mean total concentrations seem to not correspond to the distribution seen in Figure 5.
**Response**: thank you very much for your careful checking. Indeed, we have found that PAHs did not match the concentration data. We have checked our original dataset and have revised all tables. We also checked whether the data used in Figures are correct as shown in the tables.

24. Figure S5. The three legends appear as Cg… correct.
**Response**: we have corrected this term and remade the graphs.

25. Improve the quality of the figures in the supplementary material. Generally, the presentation aspects of this work need to be reviewed.
**Response**: thank you for your contribution. We have revised our quality of the figures in the supplementary material according to your good suggestions. We wish you continue helping us to improve our manuscript.

[revised manuscript text omitted]

Island Wrangel

Barents Sea
East Siberian Sea
Island Bolshoy Lyakhovsky
Island Chelyuskin
Vairaray Cape
Chukchi Peninsula
Laptev Sea
Island Vaygach
Island Kolguyev
Haranasale Cape
Kara Sea
Island Bolshoy Begichev
Island Gerkules
Island Bennett
Arctic Circle
Gydan Peninsula
Taymyr Peninsula
Russia
Prevailing winds
N
NNE
NE
ENE
E
ESE
SE
SSE
S
SSW
SW
WSW
W
WNW
NW
NNW
0 5 10km
N
Sampling site for air and water

**Figure 1.**

[Figure]

[Figure]

a
Barents Sea
Kara Sea
Laptev Sea
East Siberian Sea
b
Barents Sea
Kara Sea
Laptev Sea
East Siberian Sea
c
Barents Sea
Kara Sea
Laptev Sea
East Siberian Sea
30° E
60° E
90° E
120° E
150° E
Cd
Mn
Pb
Cu
Fe
Zn
Co
Ni
Hg

**Figure 2.**

[Figure]

a
Barents Sea
Kara Sea
Laptev Sea
East Siberian Sea
60° N
30° E
60° E
90° E
120° E
150° E
820 ng m⁻²d⁻¹
Cd
Mn
Pb
Cu
Fe
Zn
Co
Ni
Hg
b
Barents Sea
Kara Sea
Laptev Sea
East Siberian Sea
60° N
30° E
60° E
90° E
120° E
150° E
690 ng m⁻²d⁻¹
Hg

**Figure 3.**

[Figure]

a
Barents Sea
Kara Sea
Laptev Sea
East Siberian Sea
60° N
30° E
60° E
90° E
120° E
150° E
4.6 ug m⁻²d⁻¹
Cd
Mn
Pb
Cu
Fe
Zn
Co
Ni
Hg
b
Barents Sea
Kara Sea
Laptev Sea
East Siberian Sea
17 ug m⁻²d⁻¹
Naphthalene
1-Methylnaphthalene
2-Methylnaphthalene
1,4,5-Trimethylnaphthalene
1,2,5,6-Tetramethylnaphthalene
Acenaphthylene
Acenaphthene
Fluorene
Dibenzothiophene
Anthracene
9-Methlyfluorene
1,7-Dimethylfluorene
9-n-Propylfluorene
2-Methyldibenzothiophene
2,4-Dimethyldibenzothiophene
2,4,7-Trimethyldibenzothiophene
3-Methylphenanthrene
1,6-Dimethylphenanthrene
1,2,9-Trimethylphenanthrene
1,2,6,9-Tetramethylphenanthrene
Fluoranthene
Pyrene
Benzo[a]anthracene
Chrysene
3-Methylchrysene
6-Ethylchrysene
1,3,6-Trimethylchrysene
Benzo[b]fluoranthene
Benzo[k]fluoranthene
Benzo[a]pyrene
Perylene
Dibenzo[a,h]anthracene
Ineno[1,2,3-cd]pyrene
Dibenzo[a,h]anthracene
Benzo[g,h,i]perylene

[Figure]

a
Barents Sea
Kara Sea
Laptev Sea
East Siberian Sea
60° N
30° E
60° E
90° E
120° E
150° E
4.6 ug m⁻²d⁻¹
Cd
Mn
Pb
Cu
Fe
Zn
Co
Ni
Hg
b
Barents Sea
Kara Sea
Laptev Sea
East Siberian Sea
60° N
17 ug m⁻²d⁻¹
30° E
60° E
90° E
120° E
150° E
naphthalene
1-methylnaphthalene
2-methylnaphthalene
1,4,5-trimethylnaphthalene
1,2,5,6-tetramethylnaphthalene
acenaphthylene
acenaphthene
fluorene
dibenzothiophene
anthracene
9-methlyfluorene
1,7-dimethylfluorene
9-n-propylfluorene
2-methyldibenzothiophene
2,4-dimethyldibenzothiophene
2,4,7-trimethyldibenzothiophene
3-methylphenanthrene
1,6-dimethylphenanthrene
1,2,9-trimethylphenanthrene
1,2,6,9-tetramethylphenanthrene
fluoranthene
pyrene
benzo(a)anthracene
chrysene
3-methylchrysene
6-ethylchrysene
1,3,6-trimethylchrysene
benzo(b)fluoranthene
benzo(k)fluoranthene
benzo(a)pyrene
perylene
dibenzo(a,h)anthracene
indeno(1,2,3-cd)pyrene
dibenzo(a,h)anthracene1
benzo(g,h,i)perylene

**Figure 4.**

[Figure]

a
Barents Sea
Kara Sea
Laptev Sea
East Siberian Sea
b
Barents Sea
Kara Sea
Laptev Sea
East Siberian Sea
8 ng m⁻³
30° E
60° E
90° E
120° E
150° E
c
Barents Sea
Kara Sea
Laptev Sea
East Siberian Sea
14 ng L⁻¹
30° E
60° E
90° E
120° E
150° E
naphthalene
1-methylnaphthalene
2-methylnaphthalene
1,4,5-trimethylnaphthalene
1,2,5,6-tetramethylnaphthalene
acenaphthylene
acenaphthene
fluorene
dibenzothiophene
anthracene
9-methlyfluorene
1,7-dimethylfluorene
9-n-propylfluorene
2-methyldibenzothiophene
2,4-dimethyldibenzothiophene
2,4,7-trimethyldibenzothiophene
3-methylphenanthrene
1,6-dimethylphenanthrene
1,2,9-trimethylphenanthrene
1,2,6,9-tetramethylphenanthrene
fluoranthene
pyrene
benzo(a)anthracene
chrysene
3-methylchrysene
6-ethylchrysene
1,3,6-trimethylchrysene
benzo(b)fluoranthene
benzo(k)fluoranthene
benzo(a)pyrene
perylene
dibenzo(a,h)anthracene
indeno(1,2,3-cd)pyrene
dibenzo(a,h)anthracenel
benzo(g,h,i)perylene

[Figure]

a
Barents Sea
Kara Sea
Laptev Sea
East Siberian Sea
20ng m⁻³
b
Barents Sea
Kara Sea
Laptev Sea
East Siberian Sea
1 ng m⁻³
c
Barents Sea
Kara Sea
Laptev Sea
East Siberian Sea
80 ng L⁻¹
60° N
30° E
60° E
90° E
120° E
150° E
Naphthalene
1-Methylnaphthalene
2-Methylnaphthalene
1,4,5-Trimethylnaphthalene
1,2,5,6-Tetramethylnaphthalene
Acenaphthylene
Acenaphthene
Fluorene
Dibenzothiophene
Anthracene
9-Methlyfluorene
1,7-Dimethylfluorene
9-n-Propylfluorene
2-Methyldibenzothiophene
2,4-Dimethyldibenzothiophene
2,4,7-Trimethyldibenzothiophene
3-Methylphenanthrene
1,6-Dimethylphenanthrene
1,2,9-Trimethylphenanthrene
1,2,6,9-Tetramethylphenanthrene
Fluoranthene
Pyrene
Benzo[a]anthracene
Chrysene
3-Methylchrysene
6-Ethylchrysene
1,3,6-Trimethylchrysene
Benzo[b]fluoranthene
Benzo[k]fluoranthene
Benzo[a]pyrene
Perylene
Dibenzo[a,h]anthracene
Ineno[1,2,3-cd]pyrene
Dibenzo[a,h]anthracene
Benzo[g,h,i]perylene

**Figure 5.**

[Figure]

a
Barents Sea
Kara Sea
Laptev Sea
East Siberian Sea
60° N
100 ng m⁻²d⁻¹
30° E
60° E
90° E
120° E
150° E
naphthalene
1-methylnaphthalene
2-methylnaphthalene
1,4,5-trimethylnaphthalene
1,2,5,6-tetramethylnaphthalene
acenaphthylene
acenaphthene
fluorene
dibenzothiophene
anthracene
9-methlyfluorene
1,7-dimethylfluorene
9-n-propylfluorene
2-methyldibenzothiophene
2,4-dimethyldibenzothiophene
2,4,7-trimethyldibenzothiophene
3-methylphenanthrene
1,6-dimethylphenanthrene
1,2,9-trimethylphenanthrene
1,2,6,9-tetramethylphenanthrene
fluoranthene
pyrene
benzo(a)anthracene
chrysene
3-methylchrysene
6-ethylchrysene
1,3,6-trimethylchrysene
benzo(b)fluoranthene
benzo(k)fluoranthene
benzo(a)pyrene
perylene
dibenzo(a,h)anthracene
indeno(1,2,3-cd)pyrene
dibenzo(a,h)anthracene1
benzo(g,h,i)perylene

[Figure]

b
Barents Sea
Kara Sea
Laptev Sea
East Siberian Sea
60° N
1500 ng m⁻²d⁻¹
30° E
60° E
90° E
120° E
150° E
dibenzothiophene
1,2,9-trimethylphenanthrene
1,2,6,9-tetramethylphenanthrene
fluoranthene

[Figure]

Barents Sea
Kara Sea
Laptev Sea
East Siberian Sea
60° N
30° E
60° E
90° E
120° E
150° E
100 ng m⁻²d⁻¹
Naphthalene
1-Methylnaphthalene
2-Methylnaphthalene
1,4,5-Trimethylnaphthalene
1,2,5,6-Tetramethylnaphthalene
Acenaphthylene
Acenaphthene
Fluorene
Dibenzothiophene
Anthracene
9-Methlyfluorene
1,7-Dimethylfluorene
9-n-Propylfluorene
2-Methyldibenzothiophene
2,4-Dimethyldibenzothiophene
2,4,7-Trimethyldibenzothiophene
3-Methylphenanthrene
1,6-Dimethylphenanthrene
1,2,9-Trimethylphenanthrene
1,2,6,9-Tetramethylphenanthrene
Fluoranthene
Pyrene
Benzo[a]anthracene
Chrysene
3-Methylchrysene
6-Ethylchrysene
1,3,6-Trimethylchrysene
Benzo[b]fluoranthene
Benzo[k]fluoranthene
Benzo[a]pyrene
Perylene
Dibenzo[a,h]anthracene
Ineno[1,2,3-cd]pyrene
Dibenzo[a,h]anthracene
Benzo[g,h,i]perylene

**Figure 6.**

a

[Figure]

60° N
30° E
60° E
90° E
120° E
150° E
120 ng m⁻²d⁻¹
3-Methylphenanthrene
1,6-Dimethylphenanthrene
1,2,9-Trimethylphenanthrene
1,2,6,9-Tetramethylphenanthrene
Fluoranthene
Pyrene
Benzo[a]anthracene
Chrysene
3-Methylchrysene
6-Ethylchrysene
1,3,6-Trimethylchrysene
Benzo[b]fluoranthene
Benzo[k]fluoranthene
Benzo[a]pyrene
Perylene

b

[Figure]

60° N
30° E
60° E
90° E
120° E
150° E
16 ng m⁻²d⁻¹
Naphthalene
1-Methylnaphthalene
2-Methylnaphthalene
1,4,5-Trimethylnaphthalene
1,2,5,6-Tetramethylnaphthalene
Acenaphthylene
Acenaphthene
Fluorene
Dibenzothiophene
Anthracene
9-Methlyfluorene
1,7-Dimethylfluorene
9-n-Propylfluorene
2-Methyldibenzothiophene
2,4-Dimethyldibenzothiophene
2,4,7-Trimethyldibenzothiophene
Dibenzo[a,h]anthracene
Indeno[1,2,3-cd]pyrene
Dibenzo[a,h]anthracene
Benzo[g,h,i]perylene

[Figure]

Barents Sea
Kara Sea
Laptev Sea
East Siberia
60° N
30° E
60° E
90° E
120° E
150° E
130 ng m⁻²d⁻¹
naphthalene
1-methylnaphthalene
2-methylnaphthalene
1,4,5-trimethylnaphthalene
1,2,5,6-tetramethylnaphthalene
acenaphthylene
acenaphthene
fluorene
dibenzothiophene
anthracene
9-methlyfluorene
1,7-dimethylfluorene
9-n-propylfluorene
2-methyldibenzothiophene
2,4-dimethyldibenzothiophene
2,4,7-trimethyldibenzothiophene
3-methylphenanthrene
1,6-dimethylphenanthrene
1,2,9-trimethylphenanthrene
1,2,6,9-tetramethylphenanthrene
fluoranthene
pyrene
benzo(a)anthracene
chrysene
3-methylchrysene
6-ethylchrysene
1,3,6-trimethylchrysene
benzo(b)fluoranthene
benzo(k)fluoranthene
benzo(a)pyrene
perylene

**Figure 7.**

[Figure]

**Figure 8.**